# Learning a Decision Tree Algorithm with Transformers

**Yufan Zhuang**                                                          *y5zhuang@ucsd.edu*
*UC San Diego*
*Microsoft Research*

**Liyuan Liu**                                                           *lucliu@microsoft.com*
*Microsoft Research*

**Chandan Singh**                                                    *chansingh@microsoft.com*
*Microsoft Research*

**Jingbo Shang**                                                         *jshang@ucsd.edu*
*UC San Diego*

**Jianfeng Gao**                                                        *jfgao@microsoft.com*
*Microsoft Research*

**Reviewed on OpenReview:** *https://openreview.net/forum?id=1Kzzm22usl*

## Abstract

Decision trees are renowned for their ability to achieve high predictive performance while remaining interpretable, especially on tabular data. Traditionally, they are constructed through recursive algorithms, where they partition the data at every node in a tree. However, identifying a good partition is challenging, as decision trees optimized for local segments may not yield global generalization. To address this, we introduce MetaTree, a transformer-based model trained via meta-learning to directly produce strong decision trees. Specifically, we fit both greedy decision trees and globally optimized decision trees on a large number of datasets, and train MetaTree to produce only the trees that achieve strong generalization performance. This training enables MetaTree to emulate these algorithms and intelligently adapt its strategy according to the context, thereby achieving superior generalization performance.

## 1 Introduction

Transformers (Vaswani et al., 2017) have demonstrated the capacity to generate accurate predictions on tasks previously deemed impossible (OpenAI, 2023; Betker et al., 2023). Despite this success, it remains unclear if they *can directly produce models* rather than predictions. In this work, we investigate whether transformers can generate binary decision trees and explore the mechanisms by which they do so. We select decision trees as they are foundational building blocks of modern machine learning and hierarchical reasoning. They achieve state-of-the-art performance across a wide range of practical applications (Grinsztajn et al., 2022) and additionally offer interpretability, often a critical consideration in high-stakes situations (Rudin, 2018; Murdoch et al., 2019).

Decision trees are traditionally constructed using algorithms based on greedy heuristics (Breiman et al., 1984; Quinlan, 1986). To overcome the bias imposed by greedy algorithms, recent work has proposed global, discrete optimization methods for fitting decision trees (Lin et al., 2020; Hu et al., 2019; Bertsimas & Dunn, 2017). While these approaches have been effective in tabular contexts, a significant challenge lies in their non-differentiability, which raises difficulties when integrating them into deep learning models. Moreover, full

---

Code available at: https://github.com/EvanZhuang/MetaTree.

Table 1: **Comparing each algorithm's single-tree performance over the 91 held-out datasets (91D) and the 13 Tree-of-prompts datasets (ToP)**, among tree depth 2, 3, and 4. MetaTree generalizes well, despite only being trained on depth-2 trees. Listed are the average rank with standard deviation over the datasets. We also report the number of times an algorithm is in the first place (champion). [†]GOSDT encounters out-of-memory (OOM) errors for tree depths greater than 2 due to the NP-hard complexity of solving for the optimal tree. We report the memory usage for GOSDT in Table 2.

| Config | | **MetaTree** | GOSDT | CART | ID3 | C4.5 |
|---|---|---|---|---|---|---|
| 91D, depth-2 | avg. rank | $\mathbf{1.34 \pm 0.84}$ | $2.68 \pm 1.20$ | $3.30 \pm 0.90$ | $3.30 \pm 0.92$ | $4.36 \pm 1.15$ |
| | champion | **72** | 13 | 2 | 2 | 2 |
| ToP, depth-2 | avg. rank | $\mathbf{1.08 \pm 0.27}$ | $2.77 \pm 1.48$ | $3.00 \pm 1.18$ | $2.77 \pm 1.25$ | $3.38 \pm 1.60$ |
| | champion | **7** | 3 | 1 | 2 | 0 |
| 91D, depth-3 | avg. rank | $\mathbf{1.25 \pm 0.64}$ | OOM[†] | $2.65 \pm 0.78$ | $2.57 \pm 0.80$ | $3.53 \pm 0.84$ |
| | champion | **76** | - | 5 | 8 | 2 |
| ToP, depth-3 | avg. rank | $\mathbf{1.31 \pm 0.82}$ | OOM[†] | $2.54 \pm 1.15$ | $2.46 \pm 1.08$ | $2.69 \pm 0.99$ |
| | champion | **9** | - | 2 | 2 | 0 |
| 91D, depth-4 | avg. rank | $\mathbf{1.25 \pm 0.67}$ | OOM[†] | $2.79 \pm 0.79$ | $2.51 \pm 0.76$ | $3.44 \pm 0.92$ |
| | champion | **76** | - | 6 | 5 | 4 |
| ToP, depth-4 | avg. rank | $\mathbf{1.15 \pm 0.36}$ | OOM[†] | $2.46 \pm 0.93$ | $2.46 \pm 1.01$ | $2.69 \pm 1.14$ |
| | champion | **9** | - | 1 | 2 | 1 |

decision tree optimization is NP-hard (Laurent & Rivest, 1976), rendering it practically infeasible to compute optimal trees with large tree depths.

In this work, we introduce MetaTree, a transformer-based model designed to construct a decision tree on a tabular dataset. MetaTree recursively performs inference to decide the splitting feature and value for each decision node (Fig. 1a). We train MetaTree to produce high-performing decision trees. Specifically, we fit both greedy decision trees and optimized decision trees on a large number of datasets. We then train MetaTree to generate the trees that demonstrated superior generalization performance (Fig. 1c). This training strategy endows MetaTree with a unique advantage: it learns not just to emulate the underlying decision tree algorithm, but also to implicitly select which algorithmic approach is more effective for a particular dataset. This adaptability adds significant flexibility to traditional methods that are bound to a single algorithmic framework. MetaTree's architecture leverages an alternating row and column attention mechanism along with a learnable absolute positional bias for the tabular representations (Fig. 1b).

MetaTree produces highly accurate trees on a large set of real-world datasets that it did not see during training, consistently outperforming traditional decision tree algorithms (Table 1 and more in Sec. 5). Further analysis shows that MetaTree's performance improvement comes from its ability to dynamically switch between a greedy or global approach depending on the context of the split (Sec. 6.1). Finally, a bias-variance analysis shows that MetaTree successfully achieves lower empirical variance than traditional decision-tree algorithms (Sec. 6.2).

## 2 Related work

**Decision trees** There is a long history of greedy methods for fitting decision trees, e.g., CART (Breiman et al., 1984), ID3 (Quinlan, 1986), or C4.5 (Quinlan, 2014). Recent work has explored fitting trees overcoming the limitations of greedy methods by using global optimization methods (Lin et al., 2020; Hu et al., 2019; Bertsimas & Dunn, 2017; Jo et al., 2023). These methods can improve performance given a fixed tree size, but often incur a prohibitively high computational cost. Other recent studies have improved trees through regularization (Agarwal et al., 2022; Chipman & McCulloch, 2000; Yıldız & Alpaydın, 2013), iterative

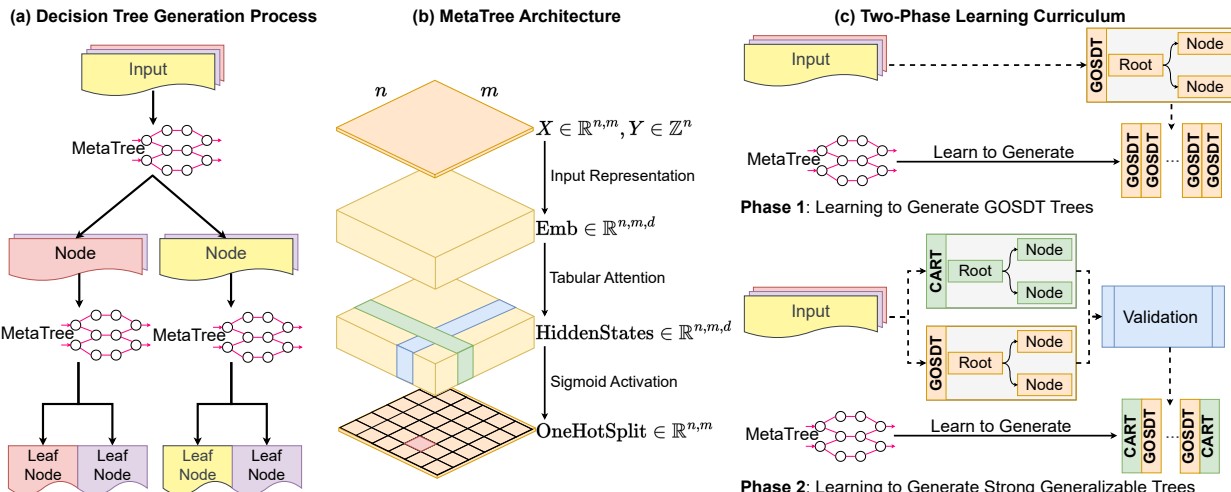

Figure 1: **MetaTree Methodology.** The creation of a decision tree, depicted in (a), entails recursive MetaTree calls. MetaTree only assesses the current state for its decision-making. (b) shows MetaTree's architecture, in which the tabular input ($n$ data points of $m$ features) is embedded in a representation space, processed with row and column attention at each layer, and the output is a one-hot mask indicating the splitting feature $j$ and threshold $X_{i,j}$. (c) illustrates MetaTree's two-phase learning curriculum: in the first phase, the focus is exclusively on learning from the optimized GOSDT trees, to closely emulate the behavior of GOSDT algorithm. Then in the second phase, the training process incorporates data from both the GOSDT and CART trees, generating the ones that have better generalization capabilities.

updates (Carreira-Perpiñán & Tavallali, 2018; Hada et al., 2023), or increased flexibility (Grossmann, 2004; Valdes et al., 2016; Tan et al., 2022; Sorokina et al., 2007; Hiabu et al., 2020).

Tree-based models maintain state-of-the-art prediction performance across most tabular applications (Grinsztajn et al., 2022; Kornblith et al., 2022), especially when used in ensembles such as Random Forest (Breiman, 2001), gradient-boosted trees (Freund et al., 1996; Chen & Guestrin, 2016), and BART (Chipman et al., 2010). Some works have sought to combine these tree-based models with deep-learning based models to improve predictive performance when fitting a single dataset (Frosst & Hinton, 2017; Zantedeschi et al., 2021; Tanno et al., 2019; Lee & Jaakkola, 2019; Biau et al., 2019).

Recent research has increasingly focused on exploring the synergy between tree-based models and large language models (LLMs). One area of investigation involves leveraging trees to guide and improve the generation processes of LLMs. Using tree structure in searching for prompts provides a hierarchical framework that structures the prompt space more effectively. It can enhance the coherence and logical flow of generated text (Yao et al., 2023), and improve task performance without needing to finetune the LLM (Morris et al., 2023). LLMs can also be used to build stronger decision trees. Aug-imodels (Singh et al., 2023) has demonstrated that integrating LLMs during the training phase of decision trees can significantly enhance their performance and interpretability.

**Learning models/algorithms** Some learning-based methods have focused on improving algorithms, largely based on combining transformers with deep reinforcement learning, e.g. faster matrix multiplication (Fawzi et al., 2022) or faster sorting algorithms (Mankowitz et al., 2023). One new work studies using LLMs to iteratively generate and refine code to discover improved solutions to problems in computer science and mathematics (Romera-Paredes et al., 2023).

Other works focus on transformers' ability to learn in context. Zhou et al. (2023) probe simple tasks for length generalization and find that transformers can generalize to a certain class of problems easily. One very related work studies whether Transformers can successfully learn to generate predictions from linear

functions, and even decision trees and small MLPs in context (Garg et al., 2022). Despite these successes, recent works also find limitations in transformers' ability to generalize in certain contexts, e.g. to distribution shifts (Yadlowsky et al., 2023) or to arithmetic changes (Dziri et al., 2023).

**Meta-learning**   Often referred to as "learning-to-learn", meta-learning has gained increasing attention in recent years (Finn et al., 2017; Nichol et al., 2018; Hospedales et al., 2021). Meta-learning algorithms are designed to learn a model that works well on an unseen dataset/task given many instances of datasets/tasks (Vilalta & Drissi, 2002). Particularly related to MetaTree are works that apply meta-learning with transformers to tabular data (Hollmann et al., 2022; Feuer et al., 2023; Onishi et al., 2023; Hegselmann et al., 2023; Gorishniy et al., 2023; Manikandan et al., 2023; Zhu et al., 2023). Our work follows the direction of meta-learning with a focus on directly outputting a model informed by the insights of established algorithms.

## 3   Methods: MetaTree

**Problem definition**   When generating a decision tree, we are given a dataset $D = (x_i, y_i)_{i=1}^{n}$ where $x_i \in \mathbb{R}^m$ represents the input features and $y_i \in \{1, \ldots, K\}$ corresponds to the label for each instance. At each node, a decision tree identifies a split consisting of a feature $j \in \{1, \ldots, m\}$ and a threshold value $v \in \mathbb{R}$, such that $D$ can be partitioned into two subsets by thresholding the value of the $j^{th}$ feature. The dataset is partitioned recursively until meeting a pre-specified stopping criterion, here a maximum tree depth. To generate a prediction from the tree, a point is passed through the tree until reaching a leaf node, where the predicted label is the majority label of training points falling into that leaf node. We illustrate MetaTree's generation pipeline, architecture design and its two-phase learning curriculum in Fig. 1.

### 3.1   Generating a decision tree model: representation and model design

Decision trees are typically fit using a top-down greedy algorithm such as CART (Breiman et al., 1984). These algorithms greedily select the split at each node based on a criterion such as the Gini impurity. This class of methods, while efficient, often results in sub-optimal solutions. More recent work has studied the generation of "optimal" decision trees seeking a solution that maximizes predictive performance subject to minimizing the total number of splits in a tree. This can be formulated as a tree search, in which recursive revisions on the tree happen when all children of a search node are proven to be non-optimal (Lin et al., 2020). However, finding optimal trees is intractable for even modest tree depths, and can also quickly overfit to noisy data.

**Model design: divide and conquer with learned speculative planning**   Our approach, MetaTree, aims to bridge the gap between existing methods by generating highly predictive decision trees, as illustrated in Fig. 1(a). To construct a single tree, MetaTree is recursively called at each node. The process begins by presenting the entire dataset to the model, in which MetaTree creates the first split (i.e. a threshold value on a feature dimension). The dataset is then filtered into two subsets by the root node's split, and each subset (i.e., left child and right child) is individually passed through the model, while masking out the opposite subset. This recursive process continues until a predetermined maximum depth for the tree is reached. Although MetaTree only outputs one split at a time, its ability to consider the entire dataset when making the initial split and utilize multiple Transformer layers enables it to make adaptive, non-greedy splits. We show experimental results supporting this claim in Sec. 6.1 and we include related case studies in Appendix A.2.

**Representing numerical inputs: learnable projection and positional bias**   MetaTree takes matrices of real-valued numbers as input. We use a multiplicative embedding to project all the numerical features into embedding space and add the class embedding onto it. The aggregated embedding is then transformed via a two-layer MLP. Specifically, given a n-row m-column input $X \in \mathbb{R}^{n,m}$ and its k-class label $Y \in \{1, \ldots, k\}^n$,

with $Y_{oh}$ denotes $Y$ in one-hot format, the embedding is computed as follows:

$$\text{(Feature Embedding)} \quad \text{Emb}_{\text{x}}(X) = X \otimes W_x \in \mathbb{R}^{n,m,d}, \quad W_x \in \mathbb{R}^d \tag{1}$$

$$\text{(Label Embedding)} \quad \text{Emb}_{\text{y}}(Y) = Y_{oh} \cdot W_y \in \mathbb{R}^{n,d}, \quad W_y \in \mathbb{R}^{k,d} \tag{2}$$

$$\text{(Position Embedding)} \quad B_{(i,j,k)} = b_{1(j,k)} + b_{2(i,k)} \tag{3}$$

$$\text{(Final Representation)} \quad \text{Emb} = \text{MLP}(\text{Emb}_{\text{x}}(X) + \text{Emb}_{\text{y}}(Y) + B) \tag{4}$$

$$\text{Emb} \in \mathbb{R}^{n,m,d}, B \in \mathbb{R}^{n,m,d}, b_1 \in \mathbb{R}^{m,d}, \ b_2 \in \mathbb{R}^{n,d}$$

For each number in the matrix $X$, it is transformed into $\mathbb{R}^d$ space via multiplication with $W_x$, then added to the class embedding of $Y$. The final embedding is obtained by putting the aggregated embedding plus the positional bias terms $b_1, b_2$ through an MLP.

We normalize each feature dimension per batch to have mean of 0 and variance of 1. Prior to inference, we also add a truncated Gaussian noise to categorical features; this improves performance on discrete features since the model is mostly trained on continuous data.

**Tabular self-attention: row and column-wise information processing**  Since our tabular input shares information across rows and columns, we apply attention to both the row dimension and column dimension in each Transformer layer. For $l^{th}$ layer, given input in hidden space $X_h^{(l)} \in \mathbb{R}^{n,m,d}$, the output of the tabular attention $Y_h^{(l)} \in \mathbb{R}^{n,m,d}$ is computed as:

$$\text{ColAttn}(X_h^{(l)}) = \text{Softmax}\left(Q_{\text{col}}^{(l)\top} K_{\text{col}}^{(l)}\right) V_{\text{col}}^{(l)} \tag{5}$$

$$\text{RowAttn}(X_h^{(l)}) = \text{Softmax}\left(Q_{\text{row}}^{(l)\top} K_{\text{row}}^{(l)}\right) V_{\text{row}}^{(l)} \tag{6}$$

$$Y_h^{(l)} = \text{ColAttn}(X_h^{(l)}) + \text{RowAttn}(X_h^{(l)}) + X_h^{(l)} \tag{7}$$

where $X_h^{(0)} = \text{Emb}(X, Y)$ for the first layer, $X_h^{(l)} = \text{MLP}(Y_h^{(l-1)})$ for the rest the layers; for the column/row query $Q$, key $K$, value $V$ projections, $X_h^{(l)} \in \mathbb{R}^{n,m,d}$ is first reshaped to $X_{h,\text{col}}^{(l)} \in \mathbb{R}^{n,m,d}$ and $X_{h,\text{row}}^{(l)} \in \mathbb{R}^{m,n,d}$ where the Softmax is conducted on the corresponding column/row dimension, *i.e.* second last dimension. The Q, K, V are constructed in the following manner, we will explain this for column attention only where the row attention follows the same design: $Q_{\text{col}}^{(l)} = W_{Q,\text{col}}^{(l)} X_{h,\text{col}}^{(l)}$, $K_{\text{col}}^{(l)} = W_{K,\text{col}}^{(l)} X_{h,\text{col}}^{(l)}$, $V_{\text{col}}^{(l)} = W_{V,\text{col}}^{(l)} X_{h,\text{col}}^{(l)}$, where the linear projections $W_{Q,\text{col}}^{(l)}, K_{\text{col}}^{(l)}, V_{\text{col}}^{(l)} \in \mathbb{R}^{d,d}$ are all learnable parameters. The dimension orders will be shuffled back after column/row attention such that $\text{ColAttn}(X_h^{(l)}), \text{RowAttn}(X_h^{(l)}) \in \mathbb{R}^{n,m,d}$.

Attention is being applied in the row and column dimensions individually, it will first gather information over $n$ rows and then over $m$ columns with an $O(n^2 + m^2)$ complexity. This alleviates the computing cost compared to reshaping the table as a long sequence, which would require an $O\left(n^2 m^2\right)$ complexity, while effectively gathering and propagating information for the entire table.

## 3.2  Training objective: cross entropy with Gaussian smoothing

The main task of our model is to select a feature and value to split the input data. This process involves selecting a specific element from the input matrix $X \in \mathbb{R}^{n,m}$. Suppose the choice is made for $X_{i,j}$, it is equivalent to a decision that splits the data along the $j^{th}$ feature with value $X_{i,j}$. Our design for the model's output and the corresponding loss function is based on this fundamental principle of split selection. The model's output is passed through a linear projection to scale down from $\mathbb{R}^{n,m,d}$ to $\mathbb{R}^{n,m,1}$, and the final output is obtained after a Sigmoid activation.

We train our model with supervised learning. In the ground truth tree, each node contains the feature index and the splitting value. This is equivalent to a one-hot mask over the input table where the optimal choice of feature and value is marked as one and the rest marked as zero. However, directly taking this mask as the training signal raises issues: some data points might have similar or exactly the same values along the same feature as the optimal split, and masking out these data points would deeply confuse the model. Hence

we use a Gaussian smoothing over this mask as our loss target based on the value distance over the chosen feature. We denote the ground truth splitting feature and value as $j^*$ and $v^*$, the training target $M$:

$$M = \begin{cases} \exp -\frac{(X[:, j]-v^*)^2}{2\sigma^2}, & \text{if } j = j^* \\ 0, & \text{if } j \neq j^* \end{cases} \qquad (8)$$

where $\sigma$ is a hyperparameter, controlling the smoothing radius. We use Binary Cross Entropy (BCE) to calculate the loss between our model output and the training target $M$.

**Learning curriculum: learning from optimized trees to best generalizing trees**  Our training approach is tailored to accommodate the mixed learning signals derived from the two distinct algorithms, each with its unique objectives and behaviors. The task is difficult. Mimicking the optimization-based decision tree algorithm is already challenging (i.e. approximating solutions for an NP-hard problem), but MetaTree must also learn to generate the split that has the better generalization potential out of the two (CART v.s. GOSDT).

To effectively train our model, we use a learning curriculum in our experiments (as illustrated in Fig. 1c. In the first phase, the focus is exclusively on learning from the optimization-based GOSDT trees. The goal during this stage is to closely emulate the behavior of GOSDT algorithm. Then in the second phase, the training process incorporates data from both the GOSDT and CART trees (see details in Sec. 4.1) to train MetaTree. This two-stage approach enables our model to assimilate the characteristics of both algorithms, facilitating better generalization capabilities. (See Appendix A.6 for ablation studies on the training curriculum.)

We selected CART and GOSDT as they are representative and effective algorithms in the classes of greedy and optimized decision tree algorithms. As shown in our experiments (Fig. 2), they are consistently first and second amongst classical algorithms. Training with the best tree out of an algorithm ensemble could potentially further improve performance, but for the simplicity of our analyses we leave that for future explorations.

**MetaTree in inference: generating decision tree models**  Note that MetaTree is equivalent to an algorithm, the inference process is slightly different from typical deep learning models. When deploying MetaTree on an unseen dataset $(X_{\text{train}}, Y_{\text{train}}, X_{\text{test}}, Y_{\text{test}})$, we will first generate decision trees from the train set $(X_{\text{train}}, Y_{\text{train}})$, then use the generated trees to make predictions $Y_{\text{pred}}$ on $X_{\text{test}}$:

$$\text{DecisionTreeModel} = \text{MetaTree}(X_{\text{train}}, Y_{\text{train}})$$
$$Y_{\text{pred}} = \text{DecisionTreeModel}(X_{\text{test}})$$

where DecisionTreeModel is constructed by recursively calling MetaTree, as shown in Fig. 1(a).

## 4  Experimental setup

### 4.1  Datasets

We use 632 classification datasets from OpenML (Vanschoren et al., 2013), Penn Machine Learning Benchmarks (Romano et al., 2021), along with a synthetic XOR dataset. We require each dataset to have at least 1000 data points, at most 256 features, at most 10 classes, and less than 100 categorical features, with no missing data. We randomly select 91 datasets as the left-out test set for evaluating our model's generalization capability while making sure they and their variants do not appear in the training set.

We generate our decision-tree training dataset in the following manner: for each dataset, we first divide it into train and test sets with a 70:30 split; then we sample 256 data points with 10 randomly selected feature dimensions from the training set and fit a GOSDT tree (Lin et al., 2020) and a CART tree (Breiman et al., 1984) of depth 2 both; we later record the accuracy of the two trees on the test set; We repeat this process and generate 10k trees for each dataset. In total, we have 10,820,000 trees generated for training.

We have both GOSDT and CART trees generated for all these datasets; for clarity, we refer to the GOSDT trees as the *GOSDT dataset*, the CART trees as the *CART dataset*, and the trees that have the best evaluation accuracy on their respective test set between CART and GOSDT as the *GOSDT+CART dataset*.

We generate a synthetic XOR dataset with the following algorithm: we first randomly sample 256 data points in the 2-dimensional bounding box $\{x | x \in [-1, 1]^2\}$; then randomly generate the ground truth XOR splits inside the bounding box depending on the pre-specified level (for example, level 1 XOR has 3 splits, level 2 XOR has 15 splits, and the root node split can take place randomly $\in [-1, 1]$ while the rest of the split is sampled randomly inside the dissected bounding boxes, see examples in Fig. A1) and assign the class labels according to the splits; at the final step, we add in label flipping noise and additional noisy feature dimensions consisting of uniform noise within $[-1, 1]$.

We additionally test MetaTree on the 13 Tree-of-prompt datasets from Morris et al. (2023). They are tabular datasets constructed from text classification tasks; the input $X$ is the LLM's response to a set of prompts accompanying the text (yes or no), and the output $Y$ is the class label. Successfully building trees on these datasets shows the potential for MetaTree to help in steering large language models. Moreover, since MetaTree is differentiable, it could be integrated into the training process of some LLMs.

### 4.2 Baselines

We use GOSDT (Lin et al., 2020), CART (Breiman et al., 1984), ID3 (Quinlan, 1986) and C4.5 (Quinlan, 2014) as our baselines, as representative optimal and greedy decision tree algorithms. For GOSDT, we utilize the official implementation, using gradient-boosted decision trees for the initial label warm-up with the number of estimators equal to 128, a regularization factor of 1e-3. For CART and ID3, we use the sklearn implementation (Pedregosa et al., 2011), setting the splitting criterion as Gini impurity and entropy respectively. For C4.5, we use the imodels implementation (Singh et al., 2021) with the default setting.

### 4.3 Model configurations

We use the LLaMA (Touvron et al., 2023) architecture as the base Transformer. For MetaTree, we set the number of layers as 12, the number of heads as 12, the embedding dimension as 768, MLP dimension as 3072. We can use a maximum of 256 data points and 10 features for producing a single tree, as Transformers' memory usage and runtime would grow quadratically with the sequence length, see our discussion on this in Sec. 7.

We pretrain our model from scratch on the *GOSDT dataset*, and after training converges, we finetune it on the *GOSDT+CART dataset*. This curriculum improves performance compared to direct training on the *GOSDT+CART dataset* (as shown in Appendix A.6). We also show an ablation with RL training objective in Appendix A.8. Detailed hyperparameters are shown in Appendix A.5.

## 5 Results: On the Generalization Ability of MetaTree

In this section, we present MetaTree's performance on real-world datasets previously unseen by the model (in Table 1 and Fig. 2). We compare it with established algorithms, GOSDT, CART, ID3, C4.5, and find that it performs favorably when used as a single tree and as an ensemble. We focus on three questions: (1) Can MetaTree effectively generalize to real-world data it has not encountered before? (2) Is MetaTree capable of generating decision trees deeper than those it was trained on?, and (3) Can MetaTree be used in an LLM setting to accurately steer model outputs?

**Generalizing to new datasets: Fig. 2A** To address the first question, we rigorously evaluate MetaTree across 91 datasets that were excluded from its training. For each dataset, we adopt the standard 70/30 split to create the train and test sets, then repeatedly sample from the train set and run the decision tree algorithms (MetaTree, GOSDT, CART, ID3, or C4.5) to form tree ensembles with specified number of trees from $\{1, 5, 10, 20, 30, 40, 50, 60, 70, 80, 90, 100\}$. The majority prediction across trees is taken as the

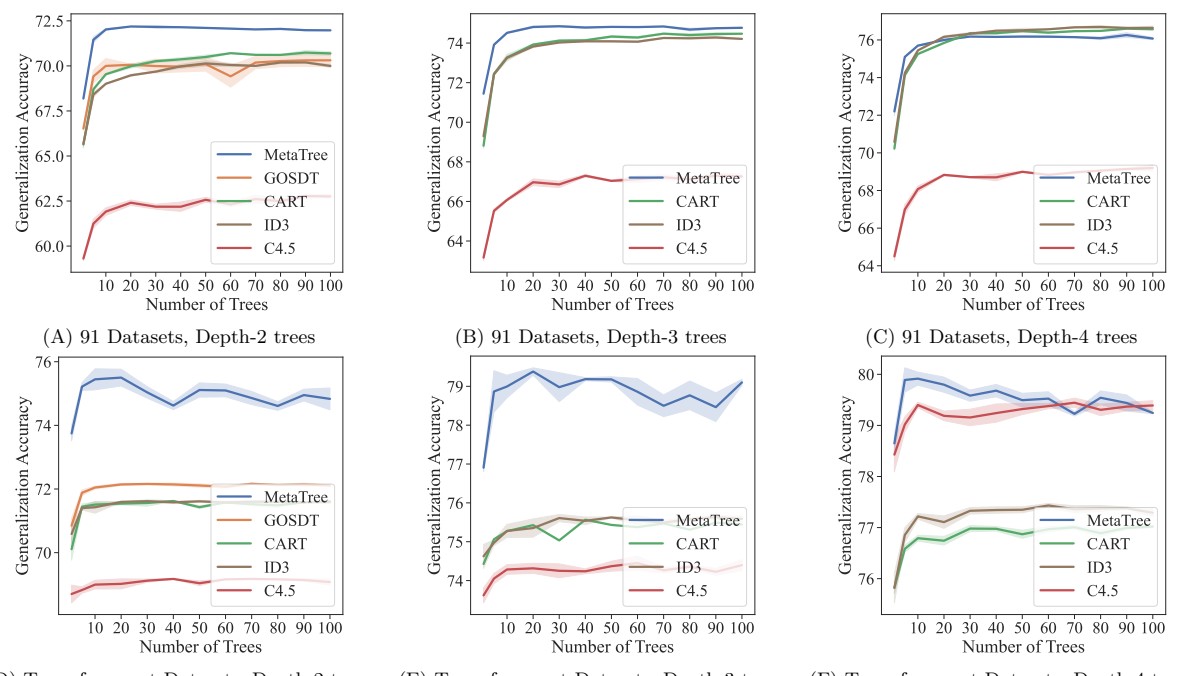

Figure 2: **MetaTree demonstrates strong generalization on real-world datasets.** MetaTree generalizes well to 91 held-out datasets for (A) depth-2 trees (B) depth-3 trees and (C) depth-4 trees, despite only being trained to produce depth-2 trees. MetaTree also generalizes well to the 13 Tree-of-prompts datasets for (D) depth-2 trees (E) depth-3 trees (F) depth-4 trees, which requires constructing a tree to steer a large language model (Morris et al., 2023). Each plot shows the average test accuracy for tree ensembles of size {1, 5, 10, 20, 30, 40, 50, 60, 70, 80, 90, 100}, with error bars indicating the standard deviation.

ensemble prediction and accuracy is averaged across all datasets. The entire evaluation process is replicated across 5 independent runs and the standard deviation is shown as error bars in the plot.

The result shows that MetaTree demonstrates a consistent and significant performance advantage over the baseline algorithms. GOSDT outperforms CART and the other greedy algorithms when the number of trees is small; this corresponds well to GOSDT's tendency to overfit on training data as it is designed to optimally solve for the train set. It also brings a higher variance in GOSDT's generalization performance. CART and ID3 exhibit similar performance due to their algorithmic similarities. In contrast, C4.5 underperforms, likely due to excessive node pruning which hampers performance when model complexity is not high enough.

In addition to the above results, we provide the average single-tree fitting accuracy of the algorithms in Appendix A.4 for a more comprehensive view of their performance.

**Generalizing to deeper trees: Fig. 2B,C** Going beyond MetaTree's capability to generalize to new data, we now study whether MetaTree can generate deeper trees. To answer this question, we ask MetaTree to generate trees with depth 3 and 4, and compare its generalization performance with the baseline algorithms, similar to the aforementioned evaluation process. [1]

The result is shown in Fig. 2B,C. It can be observed that MetaTree still consistently outperforms the baseline algorithms at depth 3. At depth 4, MetaTree outperforms the baselines in the single-tree scenario or when the number of trees is small, and the greedy algorithms (CART, ID3) catch up with the growing ensemble size.

---

[1]Note that GOSDT can not stably generate trees with a depth greater than 2 without incurring Out-of-Memory or Out-of-Time errors on machines with up to 125G memory (see Sec. 6.3 for memory usage analysis). Hence GOSDT is excluded from this study.

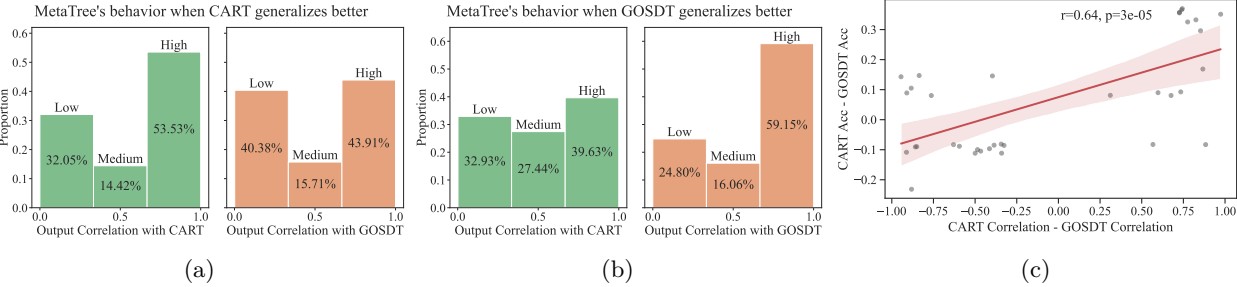

Figure 3: **We show that MetaTree learns to adapt its splitting strategy for better generalization.** (a), (b) The figures demonstrate MetaTree's tendency to select the more effective generalization strategy: opting for the greedy algorithm CART when CART performs better and the optimal algorithm GOSDT when GOSDT is the better choice. (c) We also conduct regression analysis showing that MetaTree's algorithmic preference positively correlates with the better generalizing algorithm's performance.

One reason MetaTree can generalize to deeper trees is that it is designed to generate splitting decisions at each node, hence the generated tree is not dependent on the tree depth. Besides, the model has trained on depth 2 trees, i.e. the root node split and two children split, we believe MetaTree might have learned to behave as an induction algorithm, thereby generating deeper trees with high quality.

**Tree of prompts dataset: Fig. 2D,E,F** We evaluate MetaTree on the 13 Tree-of-prompt datasets, that consist of purely categorical features, with the input being LLM's answer to a set of prompts (yes or no) and output being the classification label of the text. Fig. 2D,E,F again compares MetaTree to GOSDT, CART, ID3, and C4.5. MetaTree maintains a higher level of generalization accuracy across almost all tree counts when compared to GOSDT, CART, ID3 and C4.5.

GOSDT, CART, and ID3 show lower generalization accuracies, with GOSDT performing slightly better than CART and ID3. Notably, C4.5's performance jumps at depth-4 and surpasses CART and ID3, potentially due to its node pruning being effective at deeper tree depth.

This evaluation demonstrates MetaTree's great performance on Tree-of-prompt datasets, highlighting its capability to process categorical features and produce differentiable trees for LLM-generated inputs and user-queried outputs.

## 6 Analysis

After benchmarking the performance of MetaTree, we conduct an in-depth analysis of its behavior and splitting strategy. Our analysis of MetaTree begins by examining MetaTree's tendency between choosing a greedy split and an optimized split (Sec. 6.1). We measure the similarity of the generated splits between MetaTree and CART/GOSDT and evaluate all the splits' generalization performance. Turns out MetaTree would opt in for the better algorithm at the right circumstance. We then take a closer look at MetaTree's empirical bias-variance (Sec. 6.2). MetaTree illustrates a possibility to reduce empirical variance while not trading off empirical bias. We have included further analysis of evaluating MetaTree on synthetic XOR problems, considering the effects of noise and feature interactions (Appendix A.1). And a look into MetaTree's internal decision-making process (Appendix A.2) reveals MetaTree sometimes gets the right split early on, within the first 9 transformer layers.

### 6.1 Can MetaTree be less greedy when needed?

Our model is trained on mixed learning signals from GOSDT and CART, selected using generalization performance as the criterion. Our objective is to decipher whether MetaTree can strategically adapt its splitting approach between GOSDT and CART. To answer this question, we randomly take 100 samples (each of 256 data points) per dataset from the 91 left-out datasets and instruct MetaTree, GOSDT, and

CART to generate splits for each sample. This approach ensures that all three algorithms are provided with the same data when making the splitting decision, allowing for a meaningful comparison.

We assess the similarity between the splits generated by MetaTree and those produced by GOSDT and CART. To measure it, we calculate the correlation coefficient between the label assignments following the splits. Additionally, we evaluate the generalization performance by examining the accuracy of the splits on its respective test set. We exclude samples where GOSDT and CART yield highly similar splits (correlation coefficient > 0.7).

We plot the output correlation between MetaTree and CART/GOSDT in Fig. 3a, for when CART is the better generalizing algorithm among the two. Similarly, we plot the output correlation between MetaTree and CART/GOSDT in Fig. 3b for when GOSDT is better generalizing. We divide the output correlation values into three categories (low, medium, and high correlation), and it can be observed that MetaTree tends to favor the algorithm that exhibits better generalization performance.

Furthermore, we visualize the relationship between the correlation difference (CART correlation minus GOSDT correlation) and the generalization performance difference (CART's test accuracy minus GOSDT's test accuracy) in Fig. 3c, noting that we exclude samples with marginal performance differences (generalization accuracy difference $\leq 0.08$). A medium correlation (Pearson correlation $= 0.64$, p-value $= 0.00003$) is observed, suggesting a tendency in MetaTree's splitting strategy to align with generalization performance.

To better explain why MetaTree can learn to be non-greedy, at a high level, the model makes one decision at each node: shall I make the greedy split now? or is it better to plan for a split one step later? Due to our training strategy, MetaTree is optimized towards making these choices.

## 6.2 Breaking through the bias-variance frontier

Finally, we conduct a comprehensive bias-variance analysis to evaluate MetaTree, along with GOSDT and CART, on the 91 left-out datasets. This analysis shows how each algorithm navigates the trade-off between bias (the error due to insufficient learning power of the algorithm or incorrect model assumptions) and variance (the error due to sensitivity to small fluctuations in the training set). We use the empirical bias and variance as the measures in our evaluation; we perform 100 repetitions (N=100) for each dataset, therefore we have 100 decision tree models per algorithm per dataset. The empirical bias of an algorithm is calculated as the $\ell_2$ difference between its produced models' average output and the ground truth labels, whereas the empirical variance is calculated as the mean $\ell_2$ difference between its produced models' average output and each model's output.

The result is presented in Fig. 4a. Each point in the plot corresponds to the bias-variance coordinates derived from the performance of one algorithm on a single dataset. The x-axis represents empirical bias, indicating the algorithm's average error from the true function, while the y-axis corresponds to empirical variance, reflecting the sensitivity of the algorithm to different training sets. It can be observed that MetaTree demonstrates lower empirical variance, suggesting its robustness in diverse data distribution scenarios.

We accompany the bias-variance analysis with a dataset-level performance comparison for single trees of depth 2 in Fig. 4b. Each vertical axis represents a dataset. We first take the average generalization accuracy of CART and GOSDT as the base accuracy, then calculate the delta difference between each algorithm's accuracy and the base accuracy. Fig. 4b further illustrates MetaTree's advantage when comparing against GOSDT and CART.

## 6.3 Memory Usage and Runtime Analysis

We conduct an analysis to show the memory usage and runtime difference among CART — an efficient algorithm with greedy heuristics and GOSDT — a mathematically guaranteed optimized algorithm and MetaTree— an transformer-based algorithm that learns from both worlds.

MetaTree enjoys the benefit of a non-recursive greedy algorithm — constant memory usage and fast inference speed, while having the superior performance among them all.

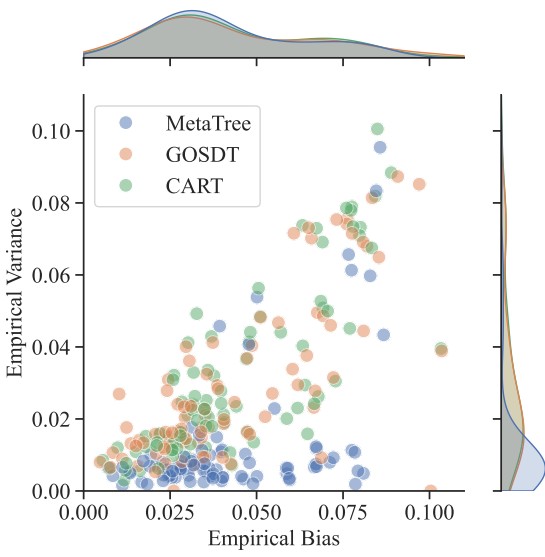
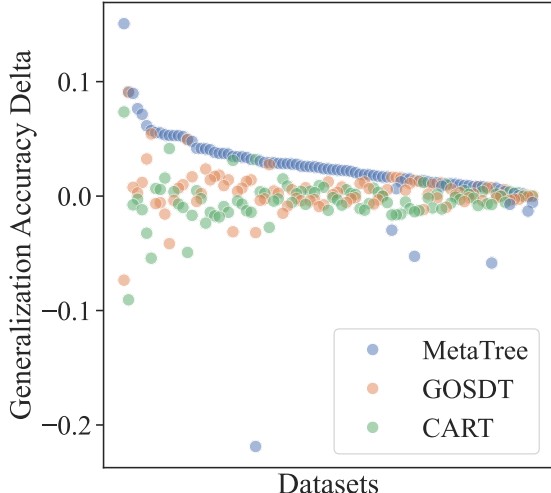

(a) Empirical bias-variance analysis.

(b) Dataset level comparisons for single trees.

Figure 4: **Empirical bias-variance decomposition for MetaTree, GOSDT, and CART** on 91 left-out datasets with 100 repetitions is shown in (a). MetaTree has significantly lower variance and slightly smaller bias as compared to GOSDT and CART. (b) We compare accuracy delta (y-axis) for each dataset (x-axis) when fitting a single tree. The delta is the change compared to the mean accuracy of CART and GOSDT.

Table 2: We show the memory usage of CART, GOSDT and MetaTree when fitting decision trees with depth 2 & 3, using the memory profiler tool (Pedregosa & Gervais, 2021). GOSDT takes a significant amount of memory since it is solving for the full decision tree solution on the data. *We report successfully terminated runs' stats, the experiments are conducted on a workstation with 125G memories and 128 cores.

|  |  | CART | GOSDT | MetaTree (GPU) |
|---|---|---|---|---|
| **Depth=2** | Mean Memory | 0.82 MB | 146.82 MB* | 2420.48 MB |
|  | Max Memory | 1.41 MB | 228.98 MB* | 2464.87 MB |
|  | Fail Rate | 0% | 1.1% | 0% |
| **Depth=3** | Mean Memory | 0.79 MB | 15,615.69 MB* | 2423.96 MB |
|  | Max Memory | 1.34 MB | 87,721.02 MB* | 2466.87 MB |
|  | Fail Rate | 0% | 86.81% | 0% |

Table 3: We show the inference wall time for CART, GOSDT and MetaTree when fitting decision trees with depth 2. GOSDT takes a significant amount of time since it is solving for the full decision tree solution on the data. *We report successfully terminated runs' stats only.

|  |  | CART | GOSDT | MetaTree (GPU) |
|---|---|---|---|---|
| **Depth=2** | Mean Time | 0.047s | 26.44s* | 0.090s |
|  | Max Time | 0.15s | 74.42s* | 0.102s |

## 6.4 Ablation studies

We conduct ablation studies on our learning curriculum (two-phase v.s. single-phase, Appendix A.6), learning signals (noisy real-world labels v.s. relabelled perfect synthetic labels, Appendix A.7), training methodology (supervised training v.s. reinforcement learning, Appendix A.8), positional bias (with positional bias v.s.

Table 4: We conducted ablation studies on various important design choices, including training curriculum (two-phase versus single-phase, Appendix A.6), learning signals (noisy real-world labels versus relabelled perfect synthetic labels, Appendix A.7), training methodology (supervised training versus reinforcement learning, Appendix A.8), positional bias (with positional bias versus without, Appendix A.9) and our choice of Gaussian smoothing hyper-parameter. We summarize the mean and std of the test accuracy changes for evaluation on the 91 left-out datasets with 100 trees.

| Ablation Setting | Performance Change |
|---|---|
| Curriculum: two-phase $\rightarrow$ single-phase | test acc $\downarrow$ $-0.79 \pm 0.14$ |
| Labels: noisy real-world $\rightarrow$ perfect synthetic | test acc $\downarrow$ $-2.78 \pm 0.23$ |
| Training: supervised $\rightarrow$ RL | training destabilized |
| Positional bias: with $\rightarrow$ without | test acc $\downarrow$ $-0.8736 \pm 0.04$ |
| Larger Gaussian radius: $\sigma = 0.05 \rightarrow \sigma = 0.1$ | test acc $\downarrow$ $-0.19 \pm 0.06$ |
| Smaller Gaussian radius: $\sigma = 0.05 \rightarrow \sigma = 0.01$ | test acc $\downarrow$ $-0.33 \pm 0.10$ |

without, Appendix A.9) and our choice of Gaussian smoothing hyper-parameter in Appendix A.10. The summary of our ablation studies is put into Table 4 and the details for each ablation experiment can be found in the corresponding section.

## 7 Discussion

**Limitations and future directions** MetaTree is constrained by the inherent architectural limitations of transformers. Specifically, the maximum number of data points and features that MetaTree can process is bounded by the transformer model's max sequence length (refer to Table A2 for detailed specifications). However, these constraints can be alleviated by training a larger model. transformers' capacity for handling long sequences is constantly improving, with state-of-the-art LLMs (Reid et al., 2024) now able to take in sequences with up to 10M tokens. While MetaTree remains limited to small datasets, this work shows an important first step in learning to adaptively produce machine-learning models, and we leave training a large-scale LLM for future work.

**Conclusion** We introduce MetaTree, a novel transformer-based decision tree algorithm. It diverges from traditional heuristic-based or optimization-based decision tree algorithms, leveraging the learning capabilities of transformers to generate strong decision tree models. MetaTree is trained using data from classical decision tree algorithms and exhibits a unique ability to adapt its strategy to the dataset context, thus achieving superior generalization performance. The model demonstrates its efficacy on unseen real-world datasets and can generalize to generate deeper trees. We conducted a thorough analysis of MetaTree's behavior and its bias-variance characteristics. More exploratory analysis and ablation studies are included in the appendix.

This work showcases the potential of deep learning models in algorithm generation, broadening their scope beyond predicting labels and into the realm of automated model creation. Its ability to learn from and improve upon established algorithms opens new avenues for research in the field of machine learning.

## Acknowledgement

Our work is sponsored in part by NSF CAREER Award 2239440, NSF Proto-OKN Award 2333790, as well as generous gifts from Google, Adobe, and Teradata. Any opinions, findings, and conclusions or recommendations expressed herein are those of the authors and should not be interpreted as necessarily representing the views, either expressed or implied, of the U.S. Government. The U.S. Government is authorized to reproduce and distribute reprints for government purposes not withstanding any copyright annotation hereon.

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

# A  Appendix

## A.1  Controlled setting: noisy XOR

We evaluate the MetaTree model in a controlled setting, on XOR datasets with level=\{1,2\}, label flipping noise=\{0\%, 5\%, 10\%, 15\%, 20\%, 25\%\}, and dataset size 10k for each XOR level/noise ratio configuration. To assess performance, we employ the *relative error* metric, defined as the gap between the achieved accuracy and the maximum possible accuracy achievable ($= 100\% -$ label noise rate). Note that we have only trained our model on 10k trees generated with XOR Level 1 and 15% label noise. This specific training scenario was chosen to evaluate the model's robustness towards noise and adaptability to harder problems.

As indicated in Table A1b, MetaTree demonstrates a remarkable capacity for noise resistance and generalization. Once MetaTree learns to solve the XOR Level 1 problem, it can withstand much stronger data noise (while MetaTree has only seen 15% noise), and generalize to significantly harder XOR Level 2 problems (while MetaTree has only trained on XOR Level 1).

We further conduct a qualitative analysis, asking CART and MetaTree to generate decision trees for XOR Level 1&2 problems. The resulting trees, as depicted in Figure A1, offer insightful comparisons into the decision-making processes of both models under varying complexity levels.

## A.2  Probing MetaTree's decision-making process

Our study is partially inspired by the logit-lens behavior analysis on GPT-2 (Hanna et al., 2023). Their findings suggest that GPT-2 often forms an initial guess about the next token in its middle layers, with subsequent layers refining this guess for the final generation distribution. Building on this concept, we aim to investigate whether a similar guessing-refining pattern exists in our model and explore the feasibility of implementing an early exiting strategy as outlined in Zhou et al. (2020).

To this end, we analyze the decision-making process of our model at each Transformer layer. We can scrutinize how the model's decisions evolve by feeding the intermediate representation from each layer into the output module. One qualitative example is shown in Fig. A1c, where we ask MetaTree to generate the root node split on an XOR Level 1 problem. We can observe that the model gets a reasonable split right after the first layer. At layer 9, the model reaches the split that is close to its final output, and layer 10's output is an alternative revision with a close to ground truth split (the ground truth can be a vertical or horizontal split).

We proceed with a quantitative analysis to investigate the correlation between the model's final split and the splits occurring in its intermediate layers. We take samples from the 91 left-out datasets following the same procedure as detailed in Sec. 6.1, and we ask MetaTree to generate splits on them. The correlation between splits is determined by calculating the correlation coefficient between the label assignments after applying the splits. This metric essentially measures how closely aligned the splits are in dissecting the input regions.

The results of our quantitative analysis are presented in Fig. A1d. Notably, the correlation shows a gradual increase from layer 1 to 8, nearly reaching 1 at layer 9. However, it then drops significantly to approximately 0.2 at layers 10 and 11. This pattern suggests that the model consistently improves its ability to make accurate predictions in the initial 1 to 9 layers, while layers 10 and 11 may introduce some divergence or revision in its decision-making process.

This finding provides valuable insights into the internal decision-making process of MetaTree. It raises the possibility of considering early exit strategies at intermediate layers, particularly around layer 9, to enhance overall efficiency.

## A.3  Statistical Comparisons for Single-trees

We conduct Fredman-Holm test Demšar (2006) to compare the single-tree level performance of MetaTree. Results are shown in Table A1. The statistical tests confirm MetaTree's performance advantage when compared against the baseline algorithms.

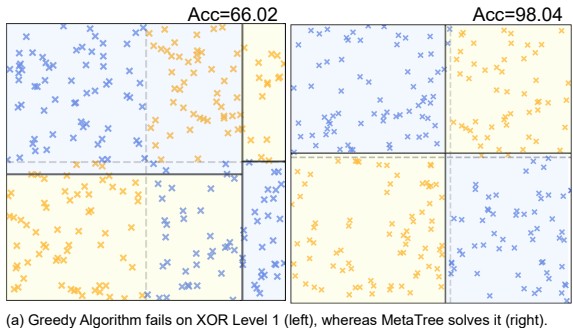

(a) Greedy Algorithm fails on XOR Level 1 (left), whereas MetaTree solves it (right).

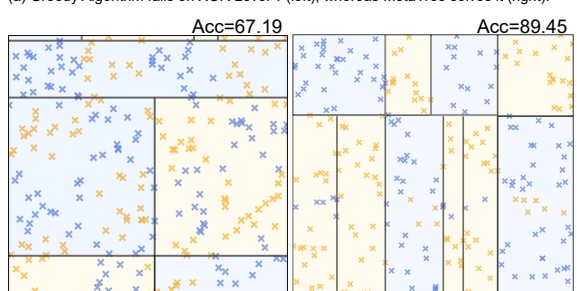

(b) Greedy Algorithm fails on XOR Level 2 (left), while MetaTree generalizes to it (right).

(a) Greedy algorithms such as CART cannot solve problems that require planning, such as Level 1&2 XOR. We show MetaTree can learn to solve Level 1 XOR and even generalize to solving Level 2 XOR.

| Noise | 0% | 5% | 10% | 15% | 20% | 25% |
|---|---|---|---|---|---|---|
| XOR L1 | 3.59 | 3.26 | 2.94 | 2.64 | 2.37 | 2.04 |
| XOR L2 | 12.78 | 11.06 | 9.34 | 7.32 | 4.99 | 2.05 |

(b) Relative error of MetaTree (trained on XOR Level 1 with 15% noise) on XOR datasets with level=$\{1, 2\}$ and label flipping noise rate from 0% to 25%.

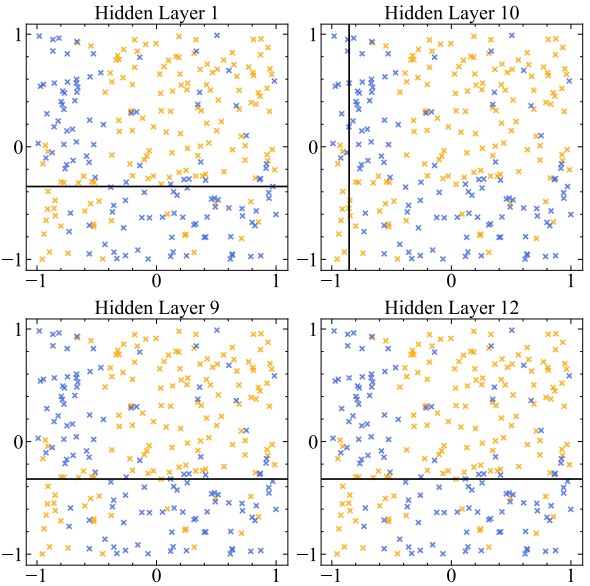

(c) Logit-lens probing of MetaTree on an XOR Level 1 problem.

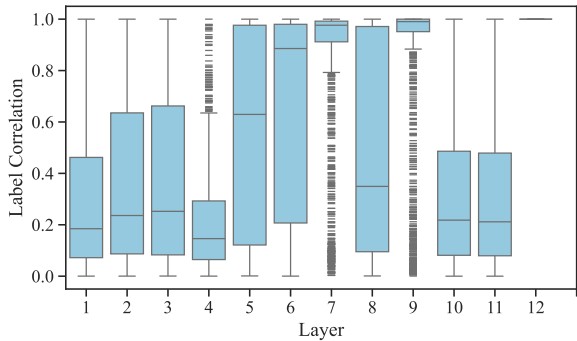

(d) The output correlation analysis between the final split and each layer's splits of MetaTree across 91 left-out datasets.

Figure A1: **Exploratory analysis of MetaTree.** (a), (b) We examined MetaTree's performance in a controlled XOR setting with various noise levels and problem difficulty. We show an illustration of MetaTree solving Level 1 XOR and generalizing to Level 2 XOR while greedy algorithms like CART are unable to solve these. (c), (d) We probe the decision-making process of MetaTree over the Transformer layers, we found out that MetaTree can very often generate the final split early on.

## A.4 Single-tree fitting accuracy

We show the fitting accuracy of the algorithms in Fig. A2 when generating single depth-2 decision trees over the datasets.

## A.5 Model Hyperparameters

We list MetaTree's hyperparameters in Table A2. We note that in the following ablation studies, the base and ablation models are trained with fewer steps (2M steps instead of 4M steps) due to the compute resource limit.

Table A1: Single-tree statistical comparison over the datasets with Fredman-Holm test Demšar (2006). We are showing the *p*-values of the ranked tests. They are all of statistical significance (i.e. $\leq 0.05$), demonstrating MetaTree's advantage over the baseline algorithms.

| MetaTree v.s. | MetaTree | GOSDT | CART | ID3 | C4.5 |
|---|---|---|---|---|---|
| 91D, depth=2 | - | 2.71e-08 | 8.88e-16 | 8.88e-16 | 0.00 |
| ToP, depth=2 | - | 0.025 | 0.017 | 0.022 | 0.004 |
| 91D, depth=3 | - | OOM | 9.42e-12 | 4.19e-11 | 0.00 |
| ToP, depth=3 | - | OOM | 0.014 | 0.011 | 0.028 |
| 91D, depth=4 | - | OOM | 2.86e-14 | 3.13e-10 | 0.00 |
| ToP, depth=4 | - | OOM | 0.006 | 0.006 | 0.006 |

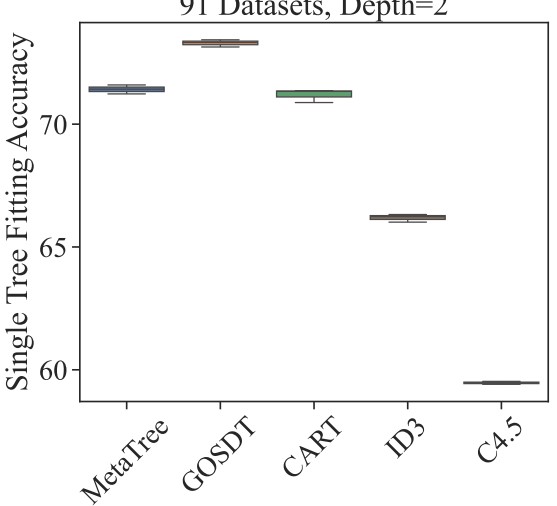

(a) Mean fitting accuracy over the 91 left-out datasets.

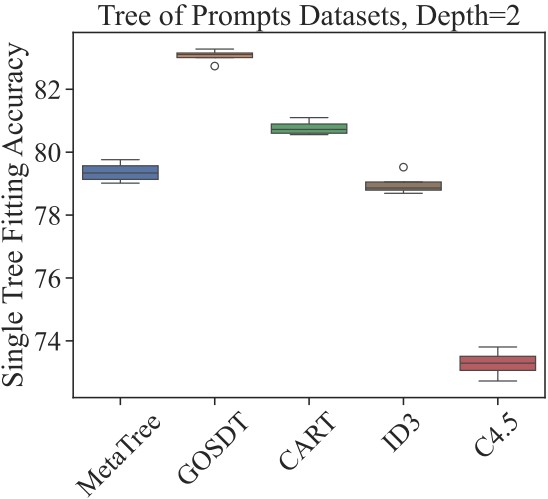

(b) Mean fitting accuracy over the 13 Tree-of-prompts datasets.

Figure A2: Average fitting accuracy for algorithms when generating single depth-2 trees.

## A.6 Ablation Study: Training Curriculum

We devise a two-phase learning curriculum for MetaTree's training, as described in Sec. 3.2. Here we conduct an ablation study comparing our curriculum with a curriculum that only uses the GOSDT+CART dataset with otherwise same configurations.

The evaluation process is the same as the one used in Sec. 5, we compute their average test accuracy on 91 left-out datasets and for the number of trees from {1, 5, 10, 20, 30, 40, 50, 60, 70, 80, 90, 100}.

As shown in Fig. A5, the learning curriculum has a significant impact on the model at the end of training.

## A.7 Ablation Study: Training Signal

We explore training with noise-free data, by relabeling the datasets with the generated trees' prediction. By relabeling the dataset, we can guarantee optimal decision splits are being fed into the model's training procedure.

We evaluate a model trained in such a manner on the 91 left-out datasets, for the number of trees from {1, 5, 10, 20, 30, 40, 50, 60, 70, 80, 90, 100}.

Table A2: Hyperparameters for MetaTree training.

| Hyperparameter | Value |
|---|---|
| Number of Hidden Layers | 12 |
| Number of Attention Heads | 12 |
| Hidden Size | 768 |
| Number of Parameters | 149M |
| Learning Rate | 5e-5 |
| Learning Rate Schedule | Linear Decay |
| Optimizer | AdamW |
| $\beta_1$ | 0.9 |
| $\beta_2$ | 0.999 |
| Training dtype | bf16 |
| Number of Features | 10 |
| Number of Classes | 10 |
| Block Size | 256 |
| Tree Depth | 2 |
| $\sigma$ | 5e-2 |
| Number of Warmup Steps | 1000 |
| Number of Training Steps | 4,000,000 |
| Steps in Phase 1 (GOSDT) | 1,000,000 |
| Steps in Phase 2 (GOSDT+CART) | 3,000,000 |
| Batch Size | 128 |

We show the results in Fig. A3, learning from the original noisy labels brings better generalization capacity on unseen datasets.

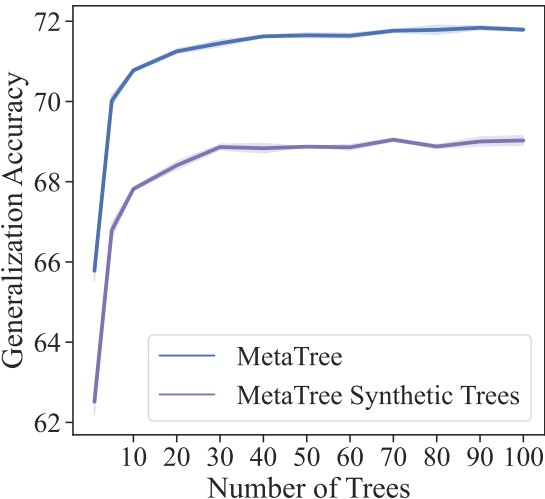

Figure A3: Average generalization accuracy for training with different labels. MetaTree trains directly from the noisy ground truth label. MetaTree Synthetic Trees is trained from the synthetic labels derived from the ground truth decision trees.

## A.8 Ablation Study: Reinforcement Learning

In the early phase of our study, We experimented with training a model directly using reinforcement learning (specifically with proximal policy optimization), where the model can explore the space of decision tree splits and get rewards as a function of the final generalization accuracy. This approach is plausible, however, we found that the training process is unstable with the sparse reward and large search space. We plot 30 runs of

such RL training in Fig. A4, where the task is to classify XOR L1 with only 2 dimensions and without any noise.

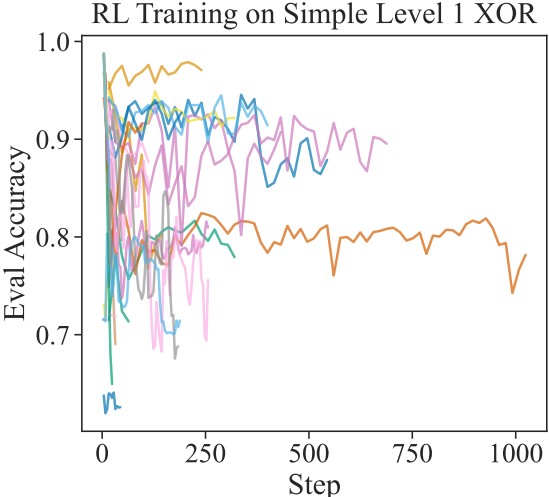

Figure A4: 30 runs of model training using reinforcement learning. It can be observed that the process is highly unstable.

As can be seen, the training process is highly unstable and rarely reaches above 95% eval accuracy, whereas the regular approach could quickly get to 97-98 eval acc. Therefore, we leave reinforcement learning for MetaTree as future research.

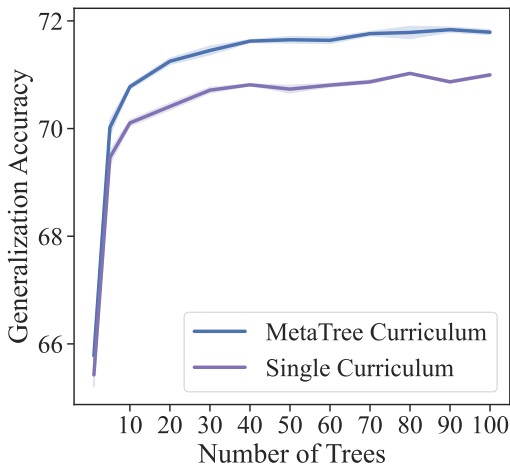

Figure A5: Average generalization accuracy for two different learning curricula. MetaTree's learning curriculum involves a first-phase training on GOSDT trees only, and a second-phase training on GOSDT+CART trees. The other curriculum is trained directly from GOSDT+CART datasets.

### A.9 Ablation Study: Positional Bias

As introduced in Sec. 3, MetaTree uses two positional bias $b_1, b_2$ at the input layer to anchor the row and column positions. We apply sequential and dimensional shuffling during training to make the model learn the positional invariance.

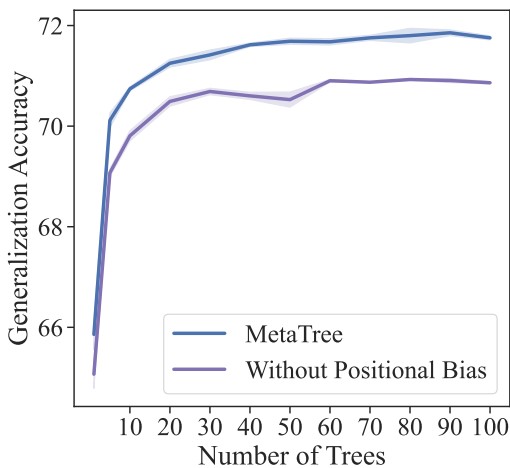

Figure A6: Average generalization accuracy for models trained with or without positional bias.

Alternatively, we can have a design that provides zero positional information to the model and becomes naturally invariant to input permutations. However, this design may have difficulties identifying the split, as the information will be mixed altogether over the Transformer layers.

We train a model with such a design, keeping all other training configurations the same, and compare its performance on the 91 left-out datasets using their average test accuracy for the number of trees from {1, 5, 10, 20, 30, 40, 50, 60, 70, 80, 90, 100}.

As shown in Fig. A6, having such positional bias is empirically beneficial compared to a design that provides no positional information to the model.

### A.10 Ablation Study: Gaussian Smoothing Loss

We conduct an ablation study on the radius of the Gaussian smoothing loss ($\sigma$). It controls how much noise and signal the model receives during training. When $\sigma$ is too large, every split may seem like a target split, but when $\sigma$ is too small, the model may fail to learn from some equally good splits.

Following the evaluation pipeline as described in Sec. 5, we conduct our ablation study comparing our model trained using $\sigma$=5e-1 with models trained with a larger $\sigma$=1e-1 and a smaller $\sigma$=1e-2, on 91 left-out datasets and compute their average test accuracy for number of trees from {1, 5, 10, 20, 30, 40, 50, 60, 70, 80, 90, 100}.

As shown in Fig. A7, the smoothing radius does have an impact on the trained model, and choosing an appropriate radius is essential to training an effective model.

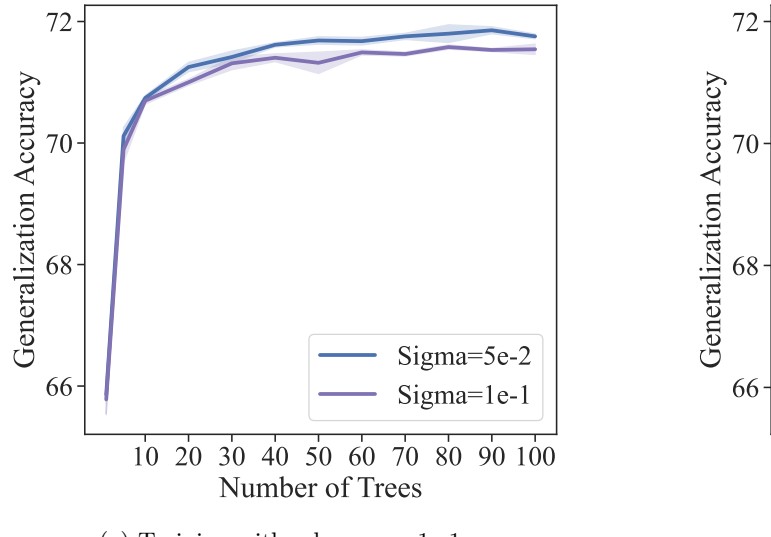
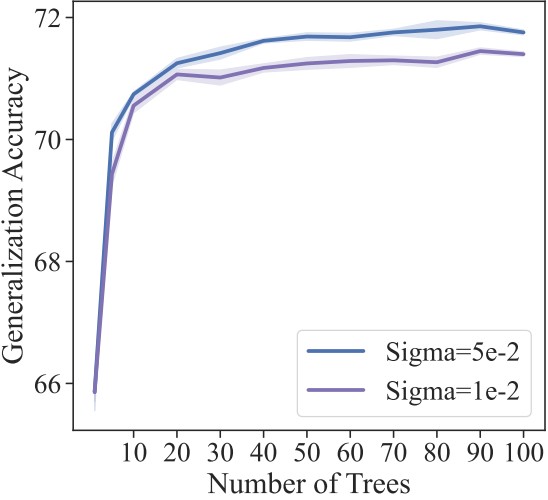

(a) Training with a larger $\sigma$=1e-1.

(b) Training with a smaller $\sigma$=1e-2.

Figure A7: Average generalization accuracy for models trained with different Gaussian Smoothing radius.

## A.11 Datasets

Table A3: List of 91 datasets used in MetaTree's evaluation.

| Dataset | Entries | Dim. | Class |
|---|---|---|---|
| mfeat fourier | 2000 | 76 | 10 |
| mfeat zernike | 2000 | 47 | 10 |
| mfeat morphological | 2000 | 6 | 10 |
| mfeat karhunen | 2000 | 64 | 10 |
| page blocks | 5473 | 10 | 5 |
| optdigits | 5620 | 64 | 10 |
| pendigits | 10992 | 16 | 10 |
| waveform 5000 | 5000 | 40 | 3 |
| Hyperplane 10 1E 3 | 1000000 | 10 | 2 |
| Hyperplane 10 1E 4 | 1000000 | 10 | 2 |
| pokerhand | 829201 | 5 | 10 |
| RandomRBF 0 0 | 1000000 | 10 | 5 |
| RandomRBF 10 1E 3 | 1000000 | 10 | 5 |
| RandomRBF 50 1E 3 | 1000000 | 10 | 5 |
| RandomRBF 10 1E 4 | 1000000 | 10 | 5 |
| RandomRBF 50 1E 4 | 1000000 | 10 | 5 |
| SEA 50 | 1000000 | 3 | 2 |
| SEA 50000 | 1000000 | 3 | 2 |
| satimage | 6430 | 36 | 6 |
| BNG labor | 1000000 | 8 | 2 |
| BNG breast w | 39366 | 9 | 2 |
| BNG mfeat karhunen | 1000000 | 64 | 10 |
| BNG bridges version1 | 1000000 | 3 | 6 |
| BNG mfeat zernike | 1000000 | 47 | 10 |
| BNG cmc | 55296 | 2 | 3 |
| BNG colic ORIG | 1000000 | 7 | 2 |
| BNG colic | 1000000 | 7 | 2 |
| BNG credit a | 1000000 | 6 | 2 |
| BNG page blocks | 295245 | 10 | 5 |
| BNG credit g | 1000000 | 7 | 2 |

| | | | |
|---|---|---|---|
| BNG pendigits | 1000000 | 16 | 10 |
| BNG cylinder bands | 1000000 | 18 | 2 |
| BNG dermatology | 1000000 | 1 | 6 |
| BNG sonar | 1000000 | 60 | 2 |
| BNG glass | 137781 | 9 | 7 |
| BNG heart c | 1000000 | 6 | 5 |
| BNG heart statlog | 1000000 | 13 | 2 |
| BNG vehicle | 1000000 | 18 | 4 |
| BNG hepatitis | 1000000 | 6 | 2 |
| BNG waveform 5000 | 1000000 | 40 | 3 |
| BNG zoo | 1000000 | 1 | 7 |
| vehicle sensIT | 98528 | 100 | 2 |
| UNIX user data | 9100 | 1 | 9 |
| fri c3 1000 25 | 1000 | 25 | 2 |
| rmftsa sleepdata | 1024 | 2 | 4 |
| JapaneseVowels | 9961 | 14 | 9 |
| fri c4 1000 100 | 1000 | 100 | 2 |
| abalone | 4177 | 7 | 2 |
| fri c4 1000 25 | 1000 | 25 | 2 |
| bank8FM | 8192 | 8 | 2 |
| analcatdata supreme | 4052 | 7 | 2 |
| ailerons | 13750 | 40 | 2 |
| cpu small | 8192 | 12 | 2 |
| space ga | 3107 | 6 | 2 |
| fri c1 1000 5 | 1000 | 5 | 2 |
| puma32H | 8192 | 32 | 2 |
| fri c3 1000 10 | 1000 | 10 | 2 |
| cpu act | 8192 | 21 | 2 |
| fri c4 1000 10 | 1000 | 10 | 2 |
| quake | 2178 | 3 | 2 |
| fri c4 1000 50 | 1000 | 50 | 2 |
| fri c0 1000 5 | 1000 | 5 | 2 |
| delta ailerons | 7129 | 5 | 2 |
| fri c3 1000 50 | 1000 | 50 | 2 |
| kin8nm | 8192 | 8 | 2 |
| fri c3 1000 5 | 1000 | 5 | 2 |
| puma8NH | 8192 | 8 | 2 |
| delta elevators | 9517 | 6 | 2 |
| houses | 20640 | 8 | 2 |
| bank32nh | 8192 | 32 | 2 |
| fri c1 1000 50 | 1000 | 50 | 2 |
| house 8L | 22784 | 8 | 2 |
| fri c0 1000 10 | 1000 | 10 | 2 |
| elevators | 16599 | 18 | 2 |
| wind | 6574 | 14 | 2 |
| fri c0 1000 25 | 1000 | 25 | 2 |
| fri c2 1000 50 | 1000 | 50 | 2 |
| pollen | 3848 | 5 | 2 |
| mv | 40768 | 7 | 2 |
| fried | 40768 | 10 | 2 |
| fri c2 1000 25 | 1000 | 25 | 2 |
| fri c0 1000 50 | 1000 | 50 | 2 |
| fri c1 1000 10 | 1000 | 10 | 2 |
| fri c2 1000 5 | 1000 | 5 | 2 |
| fri c2 1000 10 | 1000 | 10 | 2 |
| fri c1 1000 25 | 1000 | 25 | 2 |
| visualizing soil | 8641 | 3 | 2 |

| socmob   | 1156  | 1  | 2 |
|----------|-------|----|---|
| mozilla4 | 15545 | 5  | 2 |
| pc3      | 1563  | 37 | 2 |
| pc1      | 1109  | 21 | 2 |

