# OpenReview forum: "Learning a Decision Tree Algorithm with Transformers"
_TMLR — Accepted by TMLR_

### Review · Reviewer_2q7w · 2024-06-22

**Summary Of Contributions:**

This paper considers the problem of automatically learning decision trees. It takes a meta-learning approach, learning a Transformer model that, at each step, takes in a dataset and outputs a feature to split and the splitting value. The model is trained using cross entropy after discretizing the features' values, and a curriculum of optimal trees and then optimal trees + CART trees. Experiments are done on synthetic and real datasets, showing that the proposed algorithm MetaTree scales to deeper trees (unlike algorithms that compute the optimal tree) and outperforms heuristics like CART when trees have depth 2.

**Audience:**

Yes

**Broader Impact Concerns:**

No broader impact statement is included, and I think that this is fine since the paper proposes a general approach for learning decision trees and thus does not have direct societal impacts.

**Claims And Evidence:**

No

**Requested Changes:**

### Critical ###
- Discussion of computational burden (time, GPUs used, etc.) for all algorithms in the experiments and the trade-offs to be made between MetaTree and the other algorithms.

### Others ###
- A figure illustrating the model architecture.

**Strengths And Weaknesses:**

### Strengths ###
- The work is well motivated and the problem is important. Decision trees are in wide practical use today but many scalable methods are heuristic in nature. Meta-learning decision trees seems like a promising alternative approach.
- The proposed algorithm and experimental setup seems to be sound.
- Experiments are done on both synthetic and real datasets, and ablations are carried out on certain aspects of the algorithm, such as curriculum and training loss.
- Experimental results show that MetaTree outperforms both heuristics and optimal methods in terms of generalization accuracy for trees of depth 2 (trees of depth 2 were used for training).

### Weaknesses ###
- The architecture (section 3.0) is not explained very clearly. It would be helpful to add a figure.
- No discussion of the computational burden is included in the paper, although it could be one of the main advantages of MetaTree. Appendix A.11 discusses the memory usage of GOSDT and CART; I think it would be helpful to also include MetaTree and computation time.
- Once the trees get to depth 4, the generalization accuracy of MetaTree is comparable to those of other algorithms. This negatively affects the potential generalizability of the model and makes a more detailed discussion of the computational burden more important.

---

> ### Author Response · Authors · 2024-07-09
>
> We appreciate your detailed feedback and constructive suggestions. Below, we address the identified weaknesses and the requested changes to improve the clarity and comprehensiveness of our paper.
>
> 1. **Explanation of Methodology**
>
> We have taken your feedback into consideration and updated our methodology section. Additionally, we have included a new figure to better illustrate the architecture of our model.
>
> 2. **Compute burden of MetaTree**
>
> To provide a clearer understanding of the computational demands, we have conducted a detailed Runtime and Memory analysis for MetaTree, which is now available in Section 6.3 – Memory Usage and Runtime Analysis.
>
> 3. **Generalizing to deeper tree depth**
>
> We acknowledge that MetaTree's performance advantage decreases with larger tree depths. We hypothesize that training with deeper trees (beyond the current depth of 2 used in our training set) and utilizing a larger model (beyond the current 12 layers) could mitigate this issue. In this study, we have demonstrated the potential of training a transformer-based decision tree algorithm. However, similar to the evolution from BERT to GPT-4, significant advancements do not occur overnight. Therefore, we leave scaling up MetaTree for future research.
>
>
> We hope these revisions address your concerns. Should you have further questions or need additional modifications, we are ready to respond promptly. Thank you for your valuable feedback.

---

### Review · Reviewer_YdWW · 2024-06-25

**Summary Of Contributions:**

This paper propose a method to learn a Transformer based neural network MetaTree to automatically generate decision trees from a new tabular dataset. The authors develop a two-stage training curriculum to learn a highly capable and generalizable MetaTree Transformers for generating a decision tree. The authors have conducted extensive experimental evaluation to demonstrate the effectiveness of the proposed approach MetaTree.

**Audience:**

Yes

**Broader Impact Concerns:**

I do not have concerns regarding the broader impacts of this paper.

**Claims And Evidence:**

Yes

**Requested Changes:**

I would like to see more discussions on the design of the learning curriculum. It would be better to add some ablation analysis on why the proposed learning curriculum is superior.

**Strengths And Weaknesses:**

Strengths:
1. The paper is well-written. I like that the authors presented the method in Section 3 in a clear  and concise way.


2. Machine learning on tabular data is a practical yet understudied area in machine learning. I think studying how Transformers that have been shown effective in NLP and Vision generalize to other modality is an interesting and important direction.


3. The experiment and analysis conducted in the paper is thorough. I think the additional analysis in Section 6 provide a deep insight into why and how MetaTree works for generating decision trees.




Weaknesses:
1. In Figure 1c, the first stage of the training process is to emulate the GOSDT decision trees while the second stage incorporates the CART trees. Have the authors tried other training curriculum with other decision tree algorithm, e.g., ID3 and C4.5?


2. It seems not clear to me the reason for using a two-stage learning curriculum. What is the benefit of learning from the GOSDT trees in the first stage? I think this paper lacks discussion on this front which might make this part confusing to potential readers.


3. The authors mentioned in section 4.3 that the LLaMA architecture is used as the base architecture for MetaTree. However, I am not sure if the architecture designed for language is also fit for other modalities. Given that the dataset used in this paper is smaller compared to the data that LLaMA is trained on, I wonder if the authors have experimented with smaller Transformers.


4. I think it would be good to also show how architecture not based on self-attention would work for this task. What about the traditional 1d convolution, other attention formulation [1] or the recent SSM based module, e.g., Mamba [2]?

[1] Axial Attention in Multidimensional Transformers. Ho et al, 2019.
[2] Mamba: Linear-Time Sequence Modeling with Selective State Spaces. Gu et al, 2023.

---

> ### Author Response · Authors · 2024-07-09
>
> Thank you for your thoughtful and detailed review of our manuscript. We appreciate your positive feedback on the clarity of our presentation and the thoroughness of our experimental analysis. We also value your constructive criticisms, which have prompted us to refine our work further. Below, we address each of your concerns and describe the revisions made to our manuscript.
>
> 1. **Request for training curriculum ablations.**
>
> We have included an ablation study in Appendix (A.6) that examines the impact of the two-stage training curriculum. Additionally, for your convenience and to facilitate a comprehensive understanding of the effects of various design choices, we have summarized the key findings in Table 4 in the main text.
>
>  2. **Why 2-stage curriculum**
>
> The primary reason for this  2-stage approach is that directly learning the target function is maybe too difficult for the model. Therefore, we opted to initially train the model using a simpler, though still NP-hard, target. This preliminary step has proven to be an effective strategy for warming up the model.
>
> 3. **Questions on Model Architecture**
>
> - We trained our MetaTree model with 149M parameters across 12 layers, as detailed in Table A2, rather than using a model with 7 billion parameters.
> - Transformers as a foundation architecture has been successful in other modalities (for example, VIT's recent success on VLLMs). The key contribution of our research lies in shifting the paradigm from a straightforward input-output mapping to developing algorithms capable of generating mappings from input to output.
> - All of your suggested architectures are plausible to be used in MetaTree. There exist many transformer variants, including the ones you mentioned, and many can outperform vanilla transformers on specific benchmarks like Long Range Arena. However, the primary focus of our research is not solely on identifying the optimal architecture for decision tree generation. Instead, we are exploring how to effectively train a model to generate decision trees from scratch.
>
>
> We hope these revisions address your concerns adequately. We have carefully considered each of your comments and suggestions, and have made substantial improvements to the manuscript. We believe these modifications significantly enhance the overall quality and rigor of the paper. Should you have any further questions or require additional modifications, we are more than willing to address them promptly. Thank you for your insightful feedback and for giving us the opportunity to improve our work.

---

> ### Comment · Reviewer_YdWW · 2024-07-11
> **Response**
>
> Thank you for the changes and extra experiments.
>
> In the newly added ablation, changing from two-phase curriculum to one-phase curriculum incurs around 0.79% accuracy drop. It is not clear to me that this seems like an important distinction to justify the use of a two-phase curriculum. This seems to be my biggest concern at the moment and the corresponding discussion is missing in the main text.
>
> I think that my first question is still unanswered. Did the authors try different orders of the datasets used in the 2-stage curriculum?

---

> > ### Author Response · Authors · 2024-07-11
> >
> > A 0.79% difference might seem small, but it's actually quite significant when you compare it to the performance differences among classical algorithms, which are usually below that number.
> >
> > We added explanations for why we opt-ed in for this two-phase curriculum in "Learning curriculum" subsection. Essentially, this approach helps the model progressively learn the target strategy, starting from simpler to more complex tasks. What we're sharing here is based on empirical evidence, which is quite common in studies of curriculum learning.
> >
> > We chose not to use other algorithms in our experiments for a couple of reasons: 1. They generally don't perform as well as CART/GOSDT, and 2. Greedy algorithms tend to show similar behaviors. We've gone into more detail about this in Section 3.2 'Learning Curriculum'.

---

### Review · Reviewer_Dizq · 2024-06-26

**Summary Of Contributions:**

The paper proposes a strategy to generate decision trees by using a transformer architecture: the architecture produces a splitting of the input learning after trees which are generated by using CART and the optimal GOSDT algorithm by a learning curriculum strategy. The authors perform subsequently numerical experiments on real datasets showing that the strategy generates trees that can perform a classification task with good accuracy even beyond the depth of the tree actually used for the training phase. The work is of experimental/numerical nature, as it relies on the substitution of the application of some explicit splitting criterion in each decision tree level by the action of a transformer.

**Audience:**

Yes

**Broader Impact Concerns:**

There are no broader impact concerns on ethical implications requiring an explicit statement.

**Claims And Evidence:**

No

**Requested Changes:**

I list a brief collection of questions aside from some points raised in the *Weaknesses* paragraph that might be further clarified.
- The central paragraph on page 4 is, I think, too sketchy and relies on a standardization of the nomenclature in attention papers which needs nevertheless to be improved. For example, the authors never clarify what MPL means, nor how the query, key and value matrices are related to the input. Also, what is $X_h$ in the line before 4? (I expect $\mathrm{Emb}(X)$ although I do not understand if/why the notation has been changed). Finally, it is not clear what is happening in the only non-numbered equation after Eq 2: it appears that the MLP takes as input a sum of objects that have different dimensions...! Why do the authors need two biases in the expression that generates $\mathrm{Emb}$?
- In Eq. 6, how does the relative contribution of rows and columns for the generation of $Y_h$ have been decided? (Eg, it appears that the relative contribution of the two parts is not important in the efficacy of the algorithm).
- It would be useful to add some comments on the nature of the OpenML datasets (i.e., their content).  As a side note, what is the adopted criterion to sort datasets in Fig 4b?
- The authors claim that ```MetaTree's performance improvement comes from its ability to dynamically switch between a greedy or optimal approach depending on the context of the dataset```. It is not clear what *dynamically* means and I would advise further clarification on this point. Also related to this, have the authors observed any decay of accuracy for deep trees and, if this is the case, is this related to the adopted number of transformers layers?

**Strengths And Weaknesses:**

*Strengths* The work combines a nowadays very popular architecture, i.e., transformers, with a classical classification strategy, i.e., decision trees, hoping that the elsewhere observed generalization ability of transformers leads to better performances in the tree generation. The idea is, in particular, to benefit from the context sensitivity of the attention mechanism. Moreover, the differentiability of the architectural action with respect to its weight is another advantage. From this point of view, the paper is interesting as it might lead to a further step in the modern design of decision trees.

*Weaknesses* The paper is not well written, in my opinion, and some parts are kind of sketchy making it overll not clear. One weakness of the contribution is that it appears as a purely numerical experiment without discussion of a number of choices (for example, the size of the parameter sets or the impact of the embedding dimension) and without a theory-oriented analysis (e.g., the possible scaling of the accuracy with the size of the datasets, or the dimension, the number of hidden layers of the MLP, the number of transformers layers, etc). From what I understand, the training of the transformers relies on trees generated by different algorithms: the (expectedly relevant) impact of the nature of the training set is only marginally discussed. In this sense, the paper seems more of a (not very detailed) technical report without offering particular insights into *why* the proposed strategy is successful or how it is possible to extract optimal performances from the proposed algorithmic strategy.

---

> ### Author Response · Authors · 2024-07-09
>
> Thank you for your detailed review and insightful observations on our manuscript. We appreciate your engagement with our work and the opportunity to clarify and enhance the quality of our paper based on your feedback. Below, we address each of your concerns and describe the revisions made to our manuscript.
>
> 1. **Clarity and Detail of the Manuscript**
>
> 	We have made revisions to the manuscript according to your feedback, specifically in Section 3, to better explain the model’s architecture and computation process (Equations 4,5,6).
>
> 2. **Requested Ablation on Parameter Size**
>
> 	We acknowledge that exploring variations in the size of embeddings and the number of hidden dimensions could yield additional insights. However, the primary focus of our study is to demonstrate the feasibility of training a model-based decision algorithm using a meta-learning approach, and to highlight its superior performance compared to its predecessors.
>
>      While we agree that increasing the number of parameters and the volume of data can generally enhance model generalization—a trend supported by the scaling laws of large language models (LLMs)—we must also consider the computational resources required for such experiments. Given the significant computational demands, we have decided to forgo detailed ablations concerning hidden dimensions and number of layers in this initial study. We believe that our current findings provide valuable insights and pave the way for future research that could explore these aspects in greater depth.
>
> 3. **Miscellaneous**
>
> - We confirm that the contribution of Row and Column attention is even for both modules. The model will decide which one to rely on more.
>
> - The OpenML datasets are classification-oriented, we listed our criterion for selecting those datasets in Section 4.1. And the sorting criteria in Fig 4.b is the generalization performance of MetaTree, the specific order is for better illustrative purposes.
>
> - The “dynamic” means the model will adapt its greediness to the data being presented to it. For example, when given an XOR problem, MetaTree will be very non-greedy in the first split (See our analysis on XOR in the appendix), but greedy in the second split.
>
> - The performance advantage does become smaller on deeper trees, we believe it is related to the training data being limited to depth 2. It is not hard to imagine a model trained with more parameters, more training data, and deeper trees in its dataset following our framework, but due to the compute resource constraint, we leave scaling up MetaTree for future work.
>
> We hope these revisions address your concerns adequately. We believe that the changes made have significantly improved the manuscript, making the technical content more accessible and providing deeper insights into our proposed methods. We look forward to your further comments and hope that our manuscript is now suitable for publication.

---

### Review · Reviewer_j58L · 2024-06-27

**Summary Of Contributions:**

This work proposes a new supervised learning algorithm for generating decision trees. They designed a specific architecture based on a Transformer model that predicts the attribute and threshold from the data to split upon. Finally they demonstrate the model performance by predicting decision trees for several datasets.

**Audience:**

Yes

**Claims And Evidence:**

No

**Requested Changes:**

Besides addressing the weaknesses above, there are some minor corrections and clarifications:

In eq.3 you add b1, b2 and the first term which is mathematically invalid. Please clarify.

In Fig. 2 what are the number of trees?

**Strengths And Weaknesses:**

*Strengths* - generating decision trees (DT) in a non-greedy fashion could have a major impact on ML. As DTs are very interpretable and are extensively used for commercial tabular data this could have vast implications on data regimes where leading methods such as deep-learning are less effective.


*Weaknesses* - I have several concerns regarding the validity of the proposed method:

The technical parts of the paper are not well written. For example, the authors should make it clear what happens during inference time of (a) a new dataset (using the transformer), and (b) a new sample (using a DT generated by the transformer). Moreover, parts like the curriculum, data generation, and weighting of the two data-sources are not well-explained. A re-writing of Section 3 entirely could greatly improve the quality of the paper.

No analysis of the proposed features (the representation and self attention, the smoothing, the curriculum, etc’). Each feature could have been replaced with a simpler option, and understanding the contribution of each is crucial.

Empirical evidence - maybe because the curriculum and optimization processes are not well defined, it is not clear to me what creates the gap between the baselines and MetaTree given that it is trained via supervised learning. Another aspect I find confusing, is that the authors claim that the trees are closer to optimal (or less greedy), however without the information over the budget (size of the sub-tree) I can’t see how the decisions are not greedy by nature. For instance, given a fixed dataset, if we have one split vs. multiple splits can drastically change the decision in the root node. However, MetaTree is unaware of this budget. Finally, it would help if the authors explain why the datasets generated in the paper to create the MetaTree algorithm are a good representation of most of the datasets out there.

---

> ### Author Response · Authors · 2024-07-09
>
> Thank you for your thorough review and constructive feedback on our manuscript. We appreciate the time you invested in evaluating our work and your insightful comments, which have helped us identify areas for improvement. Below, we address each of your concerns and outline the revisions we have made to the manuscript.
>
> 1. **Clarity of Technical Content and Inference Process**
>
> You pointed out that the technical sections, particularly Section 3, were not clearly written, especially regarding the inference processes for new datasets and new samples. To address this:
> We have rewritten Section 3 to clarify the inference process. We now explicitly describe how the Transformer model is used during the inference time with a new dataset and how the decision trees generated by the Transformer are utilized when classifying new samples, with the added subsection titled "MetaTree in Inference".
>
> 2. **Analysis of Proposed Features**
>
> Your feedback highlighted a lack of analysis concerning the novel features of our model, such as representation and self-attention mechanisms, smoothing techniques, and the curriculum. In response:
> We have added ablation studies (Section 6.4) where we systematically evaluate each feature's contribution. This includes ablation studies and comparisons with simpler alternatives to justify our choices and demonstrate their efficacy.
>
> 3. **Non-Greedy Nature of Decision Making**
>
> We acknowledge your concerns regarding the empirical evidence and the clarity of the curriculum and optimization processes. To improve this:
> - We revised Section 3, to better explain our proposed algorithm.
> - We revised Section 6.1 to make it more intuitive to read, where we showed the evidence supporting our claim that MetaTree can be less greedy when needed. MetaTree is indeed unaware of the budget constraint, but by training with depth-2 optimal trees, it at least can think two steps ahead, which makes it closer to being optimal without inducing the huge compute cost that comes with optimal decision tree algorithms.
>
> 4. **Dataset Representativeness**
>
> Regarding the representativeness of the datasets used for developing MetaTree:
> - We found all the datasets that we can from OpenML and PMLB, with manual filtering and cleaning, so it shall be representative and help our model generalize to unseen datasets, supported by the large scale of the training data.
> - During our training, the features are normalized into a fixed range, to further strengthen the model’s generalizability and robustness.
>
> 5. **Technical Corrections**
> - In Equation 3, we have corrected the mathematical error you pointed out. We now provide a revised equation with appropriate terms and clarifications.
> - The number of trees denotes the size of the random forest.
>
>
> We hope that these revisions address your concerns satisfactorily. We are grateful for your suggestions, which have undoubtedly strengthened our paper. We look forward to your further comments and hope that our manuscript is now suitable for publication.

---

> > ### Comment · Reviewer_j58L · 2024-07-11
> > **Further modifications**
> >
> > I appreciate the modifications the authors have made, and the paper is going in the right direction, however, there are many technical details that are still unclear to me.
> >
> > "analysis shows that MetaTree’s performance improvement comes from its ability to dynamically switch betweena greedy or optimal approach depending on the context of the dataset":
> > Suggested edit: Change (also throughout the paper) from optimal to optimization-based. As you rightfully state, finding the optimal DT is NP hard, thus, even though you rely on an optimization-based method to find a DT, it won't be optimal (except for very specific cases where the features dimension is very small).
> >
> > Eq. 1-3: items with different dimensions are added.
> >
> > Per-decision context is unclear: if the data that is not part of the leaf is masked out, how does the model use it in its prediction? If it is taken into consideration, how do you only take the relevant labels? A good correction for that would be to specify in math notation what is to mask out data (page 4 "model design: .." paragraph). Or to have a pseudo code.
> >
> > page 4 "We normalize each feature dimension ... " paragraph should be in math notation and possibly in a new pre-processing sub-section. Also, is the data normalized before every call to MetaTree? or only once before the first call?
> >
> > Section 3.2: although I think I understand the intuition: "because similar contexts may have different labels in the data we want a soft update rule to make optimization smoother". However, if you are using BCE then theoretically you need your target to sum to 1. Not summing to 1, is like scaling every update by an unknown factor. This raises the question of which objective you are trying to optimize and what approximation you make along the way to make it feasible. Although the paper is written better than in the previous iteration, it is not enough in my opinion. I.e. What motivated you to use Gaussian smoothing? How do you select sigma (and what does it mean)? etc'.
> >
> > In the "learning curriculum" section: " but MetaTree must also learn to generate the split that has a better generalizationpotential." What makes you say that it has better generalization? also, compared to what? This is not very technical. If it's only because of the experiments, either refer to the relevant part, or better wait for the experiments section to say that.
> >
> > Same section: this entire section seems odd to me. Why these two algorithms specifically? Why not other algorithms as label generators? Why do you need this 2-stage approach? I think that "it just works" is justified but not ideal.
> >
> > The new "MetaTree in Inference" could be written more accurately, using math notation and fewer words.
> >
> > You keep repeating that the algorithm behaves in a non-greedy manner, but I still don't understand why. If the algorithm is unaware of the size of the subtree how can you hope to make the optimal decision? If you have an answer please elaborate on that in the paper itself.
> >
> > "We also show an ablation with RL training objectivein Appendix A.8. Detailed hyperparameters are shown in Appendix A.5." . For the casual reader this is confusing - what is RL, how is it related? What is the motivation of using that? if you plan on adding such content it shouldn't be this hand-wavy.
> >
> > "Can MetaTree be used in an LLMsetting to accurately steer model outputs?" To me, this question (much like the RL part above) seems too out of context with the rest of the paper. What is a LLM? How is it related? This extra content only confuses the reader in my opinion.
> >
> > The ablation studies in Section 6.4 are provided as a list without conclusions. These should be better integrated into the paper itself with the main conclusions provided in text. Specifically, for the curriculum I am missing the answer to why the curriculum helps and how it was designed?

---

> > > ### Author Response · Authors · 2024-07-11
> > >
> > > Thank you for your replies and further feedback. We've made these updates accordingly:
> > >
> > > 1. Update formulas in Equation 3
> > > 2. Update *MetaTree in Inference* Subsection
> > > 3. Replaced the usage of "optimal" to "optimization-based" for appropriate places
> > > 4. We revised the "Learning Curriculum" Section, added explanation of why we chose GOSDT and CART out of these algorithms.
> > > 5. Also in the "Learning Curriculum" Section, added explanation for what we mean by "better generalization potential"
> > > 5. We added explanation at the end of Sec 6.1 for why MetaTree can be non-greedy. We would also like to bring your attention to Appendix A1/A2 where we did analysis on Level 1 & Level 2 XOR problems, that specifically requires non-greedy splits up front.
> > >
> > > We'd like to clarify a few points:
> > > 1. The process for generating a decision tree is illustrated in Figure 1(a). We believe the figure clearly shows how a decision tree is created by splitting data and recursively calling MetaTree.
> > > 2. When we say, "We normalize each feature dimension per batch to have a mean of 0 and variance of 1," it means this normalization happens with each call. We think adding more formulas might actually lead to more confusion.
> > > 3. In Section 3.2, the BCE loss is calculated for each Sigmoid output item. It's important to note that this isn't the cross-entropy loss across the entire output matrix, so we avoid the issue of sums not equating to 1. Gaussian smoothing is used to control how far away in the X space a split is still considered effective. For example, if X=1.5 is the ideal split, a larger sigma (sigma=1) means X=1.6 will still be close to 1 in the target M, but with a smaller sigma (sigma=0.001), X=1.6 would be nearly 0. This approach helps the model recognize that a split at X=1.51 is nearly as effective as at X=1.5, which is explained in the main text.
> > > 4. The mention of RL and LLM in our main text might catch the interest of those who are exploring decision tree generation as a policy optimization problem or are in search of a differentiable tree generator.
> > > 5. We've included the motivations and conclusions for the ablation studies in the appendix. A brief summary in the main text should suffice, allowing readers to delve into the details in the Appendix if they choose.

---

> > > > ### Comment · Reviewer_j58L · 2024-07-24
> > > > **Thanks for the clarifications**
> > > >
> > > > Thank you for the additional clarifications, however I still have some unresolved issues:
> > > >
> > > > 1. Masking is still unclear, how is it defined in matrix ops?
> > > > I.e. it is still unclear to me if the actual transformer layers see data from "masked out" samples (whatever that means because it is not defined). Please explain more thoroughly.
> > > > Right now without looking at the code I won't be able to replicate your algorithm even with the same data.
> > > > See:
> > > > "Even though MetaTree only outputs a single split at a time, the fact that it can see
> > > > the entire dataset when making the root node split and use multiple Transformer layers allows it to make adaptive, non-greedy splits."
> > > >
> > > >
> > > >
> > > > 2. "shows that MetaTree’s performance improvement comes from its ability to dynamically switch between a greedy or optimization-based approach depending on the context of the dataset"
> > > > How is dynamically switch illustrated (for the entire dataset that is)? I understand if it was written that "MetaTree emulates a greedy vs. optimization-based target depending on the split context"
> > > > (as opposed to the dataset context).
> > > >
> > > > 3. It is still unclear why the change in strategy happens, firstly, because optimization approaches do know what size of tree they are expected to produce, thus they know the number of splits, whereas your algorithm is not aware of that information. It is clearly the case that it learns to partially imitate this target without full-knowledge, but in my opinion it is still not explaining why.
> > > > I'm not convinced by this:
> > > > "To better explain why MetaTree can learn to be non-greedy, at a high level, the model makes one decision at each node: shall I make the greedy split now? or is it better to plan for a split one step later? Due to our training strategy, MetaTree is optimized towards making these choices."
> > > > I understand the claim that it changes the target it is trying to emulate, however the explanation here is too speculative.
> > > >
> > > >
> > > >
> > > >
> > > >
> > > > 4. The loss function is still not clear to me, I'm not saying it doesn't work, I just don't fully understand the theoretical framework being used to find this good solution.
> > > >
> > > >
> > > >
> > > >
> > > >
> > > > 5. Curriculum target: when more than one tree algorithm is used (section 3.2 learning curriculum), what data is the model being trained on? is it (a) all trees one big dataset, (b) for every dataset select the best performing tree algorithm, or (c) another option?
> > > > Your experiments show that this is a crucial part of the algorithm which is also not explained well in the text.

---

> > > > > ### Author Response · Authors · 2024-07-24
> > > > > **Thanks for your further inquiries**
> > > > >
> > > > > Thank you for your further inquiries. Before we provide our explanations for each of your questions, we want to emphasize that while theoretical analysis is crucial in deep learning research, but at the current stage in 2024, it is challenging to answer questions like "how does GPT learn math proving?", "why learning to generate only the next token leads to some sort of intelligent AI?". The nature of our work is similar, highly empirical, both as you pointed out and as we stated. We conducted thorough empirical analysis around our claims and research methods, however deeper theoretical analysis would be future work when we have better theoretical tools to understand the inner mechanism of these large models.
> > > > >
> > > > > For your questions:
> > > > >
> > > > > 1. **We employed standard attention masks just as described in the original transformer paper.** At the root node, the entire attention mask is an all-ones matrix of size (batch_size, n_data, n_feature), and at later children nodes, the samples belonging to that child node still have 1 in the n_data dimension; otherwise, that row would be 0 in the attention mask. The inputs (as well as operations inside transformer layers) are augmented with the attention mask, allowing samples to be masked out.
> > > > >
> > > > > 2. Thank you for pointing it out, the original writing was ambiguous, **we've made update in the draft**: "shows that MetaTree’s performance improvement comes from its ability to dynamically switch between a greedy or optimization-based approach depending on the context of the split".
> > > > >
> > > > > 3. **We have empirical evidence for this claim, we refer you to our XOR experiments in Appendix A.1.** For level-2 XOR, that is a depth-4 tree problem and out of the original training scope, MetaTree is still able to solve it with accurate and robust performance (See example and performance table in Fig A.1). This suggests that the model has learned planning skills to some extent.
> > > > >
> > > > > 4. The design principle for our loss function was straightforward: if a split choice is good, the model's output should be close to 1 over there, and vice versa.
> > > > >
> > > > > 5. It's (b), for every dataset select the best performing tree algorithm. We refer you to Section 4.1 for details on our dataset construction process, it is also illustrated in Fig 1 (c).
> > > > >
> > > > > Thanks again for your constructive feedback and inquiries, let us know if you have further comments.

---

### Decision · Action_Editor_Cf4C · 2024-08-04

**Recommendation:** Accept with minor revision

**Comment:**

Overall the paper appears to have proposed a novel method with solid performance. One strength of this paper in my opinion is that the adaptation of transformers to this problem (meta-learning decision trees) seems to be non-trivial, e.g. in the ways that attention, positional encoding, etc are used.

On the other hand, I agree with the concerns of several reviewers that the presentation of the main algorithm (even after rebuttal) is not ideal and could be further improved. For example, the authors may consider further polishing Section 3.1 & 3.2 to make the description of the main algorithm clearer. The related work (in particular related to ICL and meta-learning) could also be more comprehensive. In my opinion this paper could serve as a first work in the direction of learning decision trees by attention-based architectures, and an adequate literature review could make the paper stand stronger in such a position. For example, the difference with the works in the Section "Some recent works have studied the intersection of trees and Transformers" could be discussed in more detail.

After rebuttal, most reviewers are supportive of the paper. Therefore, I recommend acceptance with minor revision, and encourage the authors to revise the paper according to the reviewers' and the above suggestions.

**Audience:**

This paper would be of interest to a good part of the general machine learning community, in particular those on transformer architectures for structured datasets, as well as learning decision trees.

**Claims And Evidence:**

This paper presents an attention-based architecture called MetaTree that can learn a decision tree from any dataset. The architecture incorporates many ingredients of transformers and adapts them to the decision tree setting, and is (meta)-learned on an 'expert' dataset produced by existing SOTA decision tree algorithms. Experiments validate the performance of MetaTree against existing algorithms.